# Molecular Manipulation of the MiR396/*GRF* Expression Module Alters the Salt Stress Response of *Arabidopsis thaliana*

Joseph L. Pegler [1], Duc Quan Nguyen [1,2], Jackson M.J. Oultram [1], Christopher P.L. Grof [1] and Andrew L. Eamens [1,3,*]

1  Centre For Plant Science, Faculty of Science, School of Environmental and Life Sciences, University of Newcastle, Callaghan, NSW 2308, Australia; joseph.pegler@newcastle.edu.au (J.L.P.); ducquan.nguyen@uon.edu.au (D.Q.N.); jackson.oultram@uon.edu.au (J.M.O.); chris.grof@newcastle.edu.au (C.P.G.)
2  Institute of Genome Research, Vietnam Academy of Research and Technology, 18 Hoang Quoc Viet Str., Cau Giay, Hanoi 100000, Vietnam
3  School of Science, Technology and Engineering, University of the Sunshine Coast, Maroochydore, QLD 4558, Australia
*  Correspondence: andy.eamens@newcastle.edu.au or aeamens@usc.edu.au; Tel.: +61-249-217-784

**Abstract:** We previously demonstrated that microRNA396 (miR396) abundance is altered in 15-day-old *Arabidopsis thaliana* (*Arabidopsis*) whole seedlings following their exposure to a 7-day salt stress treatment regime. We, therefore, used a molecular modification approach to generate two new *Arabidopsis* transformant populations with reduced (*MIM396* plants) and elevated (*MIR396* plants) miR396 abundance. The exposure of 8-day-old wild-type *Arabidopsis* whole seedlings and a representative plant line of the *MIM396* and *MIR396* transformant populations to a 7-day salt stress treatment regime revealed unique phenotypic and physiological responses to the imposed stress by unmodified wild-type *Arabidopsis* plants and the *MIM396* and *MIR396* transformat lines. A quantitative reverse transcriptase polymerase chain reaction (RT-qPCR) approach was, therefore, applied to demonstrate that the plant line specific responses to salt stress likely stemmed from the unique molecular profile of each of the *GROWTH REGULATING FACTOR* (*GRF*) transcription factor gene family members which form posttranscriptional targets of miR396-directed expression regulation. RT-qPCR additionally revealed that, in 15-day-old *Arabidopsis* whole seedlings, the three previously identified putative target genes of miR396 belonging to the *NEUTRAL/ALKALINE NONLYSOSOMAL CERAMIDASE-LIKE* (*NCER*) gene family, including *NCER1*, *NCER2*, and *NCER3*, do not form targets of miR396-directed expression regulation at the posttranscriptional level. Taken together, the phenotypic and molecular analyses presented here demonstrate that alteration of the miR396/*GRF* expression module is central to the molecular response of *Arabidopsis* to salt stress.

**Keywords:** *Arabidopsis thaliana* (*Arabidopsis*); salt stress; molecular manipulation; microRNA396 (miR396); *GROWTH REGULATING FACTOR* (*GRF*) genes; *NEUTRAL/ALKALINE NONLYSOSOMAL CERAMIDASE-LIKE* (*NCER*) genes; miRNA-directed gene expression regulation; RT-qPCR

## 1. Introduction

The plant-specific GROWTH REGULATING FACTOR (GRF) family of transcription factors was initially discovered in rice (*Oryza sativa*) with family members now characterized in the agronomically important monocotyledonous grasses, maize (*Zea mays*) and wheat (*Triticum aestivum*), as well as in the genetic model dicotyledonous plant species, *Arabidopsis* (*Arabidopsis thaliana*) [1–6]. At the posttranscriptional level, the expression of specific members of the *GRF* transcription factor gene family is regulated by the small non-coding RNA (sRNA), microRNA396 (miR396) [7–9]. Studies across multiple plant species have now documented the central role occupied by the miR396/*GRF* expression module in an array of developmental processes, including (1) stem elongation and grain development

in rice [1,10,11], (2) kernel and ear development in maize [4], and (3) cell proliferation and cell aging in *Arabidopsis* [5–7,12–14]. Taking rice as an example of the importance of the miR396/*GRF* expression module to plant development, miR396 has been shown to limit yield via repressing the expression of the rice *GRF*s, *GRF4* and *GRF6* [10,11,15]. More specifically, via the use of a molecular modification approach to liberate the *GRF4* and *GRF6* gene transcripts from miR396-directed expression regulation, elevated *GRF4* and *GRF6* transcript abundance was correlated with increased grain size and the promotion of panicle branching, which together resulted in higher yield [10,11,15].

In addition to mediating a central role in plant development, alterations to the abundance of miR396 or the expression of its *GRF* target genes have been demonstrated to form an important part of the molecular response of *Arabidopsis*, rice, maize, and wheat to numerous abiotic stress challenges, including cold, drought, heat, osmotic, salt, and UV stress [2,3,16–19]. Taking drought and salt stress as examples, reduced miR396 abundance has been reported in rice, barrelclover (*Medicago truncatula*), Emmer wheat (*Triticum dicoccoides*), and sorghum (*Sorghum bicolor*) [3,20–22]. Together, these findings strongly suggest that release of *GRF* target gene expression repression by miR396 forms a central component of the molecular response of these plant species to drought stress. Additionally, altered miR396 abundance has also been reported in salt-stressed *Arabidopsis*, cotton (*Gossypium hirsutum* L.), creeping bentgrass (*Agrostis stolonifera*), rice, tobacco (*Nicotiana tabacum*), and tomato (*Solanum lycopersicum*) [23–26]; these findings further indicate that alteration of the miR396/*GRF* expression module is required as part of the molecular response of these evolutionarily diverse plant species to this form of abiotic stress. Furthermore, the *Arabidopsis grf7* single mutant, a plant line defective in the activity of GRF7, a well-documented posttranscriptional target of miR396-directed expression regulation, displays a heightened tolerance to drought and salt stress [17,27]. The enhanced tolerance phenotype displayed by *Arabidopsis grf7* plants, when compared to wild-type *Arabidopsis* plants of the same age, likely stems from the elevated expression of a large cohort of stress-responsive genes [17,27]; this finding again highlights the central requirement of the miR396/*GRF* expression module for the molecular response of *Arabidopsis* to these specific abiotic stressors.

The overaccumulation of salt in the soil of arable land can be the result of natural events, including elevated rates of evapotranspiration and/or coastal flooding events [28–30]. In recent decades, however, human activities relating to agriculture, such as poor irrigation practices, excessive fertilizer application, and the mechanical overcultivation of soil cleared of its natural vegetative cover have greatly increased both the rate and extent of accumulation of salt ions in the soils of once productive agricultural land [28–30]. Soil salinization is now one of the most significant environmental concerns facing modern agriculture, with approximately 20% of the world's land suitable for cultivation now adversely affected by salinity [31,32]. Following the uptake of salt ions from the soil by the root system, the salt ions are further concentrated in the aboveground tissues of a plant, which in turn interfere with numerous biochemical and physiological processes [29,30,33,34]. In the major cereal crop species maize, rice, and wheat, for example, the over-accumulation of salt ions leads to significant yield reductions due to decreased water and nutrient uptake from the soil, repressed carbon assimilation, defective stomata opening, and, therefore, a greatly reduced photosynthetic capacity [29,30,33–38].

Due to a plant being restricted to its site of germination, highly intricate and interrelated molecular pathways have evolved to limit the degree of developmental progression inhibition stemming from abiotic stress exposure [34,39–42]. At the posttranscriptional level, for example, the miRNA class of sRNA has been repeatedly demonstrated to occupy a central role in coordinating the complex cascade of gene expression events required by a plant to mount a molecular response to abiotic stress challenge [28,42–45]. Therefore, to build on our previous finding that miR396 abundance is altered in 15-day-old *Arabidopsis* whole seedlings following their exposure to a 7-day salt stress treatment regime [19], we applied a molecular modification approach to develop two new *Arabidopsis* transformant populations, termed *MIM396* and *MIR396* transformants. Reduced or enhanced miR396

abundance in *MIM396* or *MIR396* plants, respectively, was revealed to have an opposing effect on the development of these two transformant lines under control growth conditions. Furthermore, the *MIM396* and *MIR396* transformant lines displayed phenotypic and physiological responses which were unique to each transformant line, and to those displayed by unmodified wild-type *Arabidopsis* whole seedlings post the exposure of the three assessed plant lines to the salt stress treatment regime. The molecular profiling of *GRF* target gene expression in control and salt-stressed *MIM396* and *MIR396* plants by quantitative reverse transcriptase polymerase chain reaction (RT-qPCR) assessment revealed that the altered development and plant line-specific phenotypic and physiological response of these two transformant lines to salt stress were the likely result of the unique combination of *GRF* target transcript abundance changes detected in each transformant line. RT-qPCR additionally revealed that, in 15-day-old *Arabidopsis* whole seedlings, the three previously identified putative target genes of miR396-directed expression regulation belonging to the *NEUTRAL/ALKALINE NONLYSOSOMAL CERAMIDASE-LIKE* (*NCER*) gene family do not form targets of miR396-directed expression regulation at the posttranscriptional level. When taken together, the phenotypic and molecular analyses presented here demonstrate that altered miR396 abundance and, therefore, modified *GRF* target gene expression form a central component of the molecular response of *Arabidopsis* to salt stress.

## 2. Materials and Methods

### 2.1. Plant Expression Vector Construction

The plant expression vectors, p*AtMIM396* and p*AtMIR396*, used to generate the *MIM396* and *MIR396* transformant populations, respectively, were produced via the placement of artificially synthesized (Integrated DNA Technologies, Australia) DNA fragments behind the *Cauliflower mosaic virus* (CaMV) 35S promoter of the pBART plant expression vector backbone. More specifically, for the construction of the p*AtMIM396* plant expression vector, the endogenous non-cleavable target site of miR399 harbored by the *Arabidopsis* non-protein-coding RNA, *INDUCED BY PHOSPHATE STARVATION1* (*IPS1*; *AT3G09922*), was replaced with a non-cleavable target site sequence specific to the miR396a sRNA according to the design method outlined in [46,47]. To generate the miR396 overexpression transgene, p*AtMIR396*, a DNA fragment was artificially synthesized to match the non-protein-coding RNA sequence of the *Arabidopsis PRE-MIR396A* (*AT2G10606*) precursor transcript. Following plasmid sequencing to confirm the integrity of the p*AtMIM396* and p*AtMIR396* plant expression vectors, both plasmid vectors were extracted from *Escherichia coli* strain DH5$\alpha$ and subsequently introduced into *Agrobacterium tumefaciens* (*Agrobacterium*) strain GV3101 in preparation for *Agrobacterium*-mediated transformation of wild-type *Arabidopsis* plants.

### 2.2. Plant Material Preparation for Agrobacterium-Mediated Transformation

Wild-type *Arabidopsis* seeds (ecotype; Columbia-0 (Col-0)) were planted out directly onto the surface of a standard soil mixture for *Arabidopsis* cultivation (Seeds and Cuttings Mix, Debco, Australia). The pots were then transferred to 4 °C and incubated for 48 hours (h) in the dark to stratify the seed to ensure uniform rates of germination and developmental progression. Post stratification, pots were transferred to a temperature-controlled growth cabinet (A1000 Growth Chamber, Conviron, Australia) and the seeds were germinated and cultivated under a standard *Arabidopsis* growth regime of a 16/8 h day/night light cycle (100–120 µmol·m$^{-2}$·s$^{-1}$) and a day/night temperature of 22/18 °C until the plants transitioned to the reproductive phase of development. At approximately 4 weeks of age, all opened flowers and/or siliques which formed on the primary inflorescence stems of Col-0 plants were manually trimmed with the remaining reproductive tissue, namely, the terminal floral bud, used for *Agrobacterium*-mediated floral dip transformation according to the protocol of [48].

Putative transformants harboring the introduced *MIM396* and *MIR396* transgenes were identified via '*plating out*' the T$_1$ seeds harvested from the '*dipped*' T$_0$ Col-0 plants. More specifically, the T$_1$ seeds were surface sterilized via a 90 minute (min) treatment period

in a sealed chamber with chloride gas. Post sterilization, seeds were spread out onto plates that contained standard *Arabidopsis* growth medium (half-strength Murashige and Skoog (MS) salts) which was supplemented with the selective agent, phosphinothricin (PPT), at a concentration of 10 mg/L. The plates were sealed with gas permeable tape and incubated at 4 °C in the dark for 48 h for stratification to ensure uniform rates of germination and developmental progression of each putative *MIM396* and *MIR396* transformant. Following 2 weeks of cultivation on selective medium, each resistant $T_1$ plant was transferred to soil to allow the plant to fully mature for the collection of $T_2$ seed. The $T_2$ generation was also exposed to the same selective process in addition to the (1) transgene copy number and (2) zygosity of each putative transformant line being determined in this generation via a standard PCR-based genotyping approach. Of the $T_2$ transformant lines determined to be homozygous for a single copy of each introduced transgene, the '*best-performing*' transformant line was selected as the representative plant line of the *MIM396* and *MIR396* transformant populations for subsequent phenotypic and molecular analyses which were conducted on the $T_3$ generation. The best-performing *MIM396* and *MIR396* plant line was selected from their respective transformant populations following RT-qPCR analysis of miR396 abundance in the $T_2$ generation, i.e., identification of the (1) *MIM396* transformant line with the greatest degree of reduced miR396 abundance, and (2) *MIR396* transformant line determined to have the highest elevation in miR396 abundance.

### 2.3. Application of the 7-Day Salt Stress Treatment Regime

Seeds sourced from Col-0 plants and those of the selected *MIM396* and *MIR396* transformant lines were surface sterilized, transferred to standard *Arabidopsis* growth medium, and stratified as outlined above. The sealed plates were transferred to a temperature-controlled growth cabinet, and the seeds were germinated and cultivated for an 8-day period under a standard *Arabidopsis* growth regime (outlined above). For the salt stress treatment regime, equal numbers ($n = 48$) of 8-day-old Col-0, *MIM396*, and *MIR396* seedlings were transferred to either (1) fresh standard *Arabidopsis* growth medium (control treatment), or (2) fresh *Arabidopsis* growth medium which was supplemented with 150 mM NaCl salt stress treatment. Post seedling transfer, the control and salt stress treatment plates were sealed with gas permeable tape and returned to the temperature-controlled growth cabinet. The control and salt-stressed Col-0, *MIM396*, and *MIR396* seedlings were cultivated for an additional 7-day period under a standard *Arabidopsis* growth regime (outlined above), and all phenotypic, physiological, and molecular assessments reported here were conducted on 15-day-old Col-0, *MIM396*, and *MIR396* whole seedlings following this 7-day growth period.

### 2.4. Phenotypic and Physiological Assessment of 15-Day-Old Control and Salt-Stressed Col-0, MIM396, and MIR396 Seedlings

The phenotypic measurements of fresh weight (mg), rosette area ($mm^2$), and primary root length (mm) were converted to a percentage to allow for direct comparison of these metrics for the newly generated *MIM396* and *MIR396* transformant lines to those of 15-day-old control grown Col-0 seedlings which were assigned a value of 100%. Rosette area was calculated via the analysis of photographic images of control and salt-stressed Col-0, *MIM396*, and *MIR396* seedlings grown on media plates which were orientated horizontally for the entire 15-day experimental period with the freely available software, ImageJ. Similarly, primary root length was determined via ImageJ analysis of photographic images of control and salt-stressed 15-day-old Col-0, *MIM396*, and *MIR396* seedlings which were grown on vertically orientated plates for the 7-day treatment period post the transfer of 8-day-old seedlings to fresh *Arabidopsis* growth medium.

The physiological parameters of anthocyanin content and chlorophyll *a* and *b* abundance were also determined for the *MIM396* and *MIR396* transformant lines for their direct comparison to the corresponding metrics calculated for 15-day-old control grown Col-0 seedlings; these values were again assigned a value of 100% for ease of comparison. To determine the anthocyanin content of each sample, 100 mg of freshly harvested rosette leaf

material was ground into a fine powder in liquid nitrogen ($LN_2$); then, once the powder thawed, 1.0 mL of acidic methanol (containing 1.0% ($v/v$) of HCl) was added to each sample, and the samples were incubated at 4 °C for 2 h. The plant debris which remained after this incubation period was pelleted out of solution via centrifugation of the samples at $15,000 \times g$ for 5 min at room temperature. The absorbance (A) of each sample was then measured in a GENESYS 10S spectrophotometer (ThermoFisher Scientific, Australia) at wavelengths 530 ($A_{530}$) and 657 ($A_{657}$) nm, using acidic methanol as the blanking solution. As outlined in [49], the anthocyanin content of each sample, as determined in the units of milligrams per gram of fresh weight (mg/g FW), was calculated using the following equation: $A_{530} - 0.25 \times A_{657}$/fresh weight (g).

For the determination of the chlorophyll *a* and *b* abundance of control and salt-stressed 15-day-old Col-0, *MIM396*, and *MIR396* seedlings, 100 mg of freshly harvested rosette leaf material was ground into a fine powder in $LN_2$ and, to this powder, 1.0 mL of 80% ($v/v$) ice-cold acetone was immediately added. The samples were briefly mixed by careful hand inversion and then incubated in the dark for 24 h at room temperature. Following this incubation period, any remaining cellular debris was pelleted out of the sample solution via centrifugation for 5 min at room temperature at $15,000 \times g$. The A of each sample was determined at wavelengths 646 ($A_{646}$) and 663 ($A_{663}$) nm in a spectrophotometer and via the use of 80% ($v/v$) acetone as the blanking solution. The abundance of chlorophyll *a* and *b* was then determined using the Lichtenthaler's equations exactly as outlined in [50], and this initially obtained value was subsequently converted to a value of micrograms per gram of fresh weight (µg/g FW).

### 2.5. RT-qPCR Assessment of miR396 Abundance and Target Gene Expression in 15-Day-Old Control and Salt-Stressed Col-0, MIM396, and MIR396 Seedlings

A standard TRIzol™ Reagent protocol (Invitrogen™, Australia) was used to extract total RNA from control and salt-stressed 15-day-old Col-0, *MIM396*, and *MIR396* seedlings. Following total RNA extraction, a NanoDrop® spectrophotometer (NanoDrop® ND-1000, Thermo Scientific, Australia) was employed to determine the concentration of each extraction. For each extraction deemed to be of appropriate concentration, a standard electrophoresis approach on a 1.2% ($w/v$) ethidium bromide-stained agarose gel was used to visualize the quality of each extraction. Four biological replicates consisting of pools of eight 15-day-old control and salt-stressed Col-0, *MIM396*, and *MIR396* whole seedlings were subsequently used as templates for the synthesis of complementary DNA (cDNA) to quantify miR396 abundance or the level of gene expression.

The synthesis of a cDNA product specific to the miR396 sRNA was conducted as previously reported in [19] using a protocol adapted from [51]. A high-molecular-weight (HMW) cDNA library was also constructed from the control and salt stress samples as previously described in [52,53]. Following the synthesis of sRNA and HMW cDNA fractions, all RT-qPCR assessments of the abundance of miR396 or the expression of its target transcripts were conducted using the same cycling conditions of (1) $1 \times 95$ °C for 10 min, followed by (2) $45 \times 95$ °C for 10 s and 60 °C for 15 s. In addition, for all performed RT-qPCR experiments, the GoTaq® qPCR Master Mix (Promega, Australia) was used as the fluorescent reagent, and, post this analysis, miR396 abundance or target transcript expression was quantified using the $2^{-\Delta\Delta CT}$ method. Furthermore, the small nucleolar RNA, *snoR101*, and the housekeeping gene, *UBIQUITIN10* (*UBI10*; *AT4G05320*), were used for the normalization of miR396 abundance and target transcript expression, respectively. Supplementary Table S1 lists the sequences of the DNA oligonucleotides used in this study for the reported RT-qPCR analyses.

### 2.6. Statistical Analysis

As stated above, the phenotypic and molecular data reported in this study were obtained from the analysis of four biological replicates. Statistical analysis was performed using the one-way analysis of variance (ANOVA; RRID:SCR_002427) method while Tukey's *post hoc* test was performed using the SPSS Program (IBM, United States;

RRID: SCR_002865). The results of these analyses are presented as letters above the columns on the relevant histograms in Figures 1–4. The same letter above a histogram column indicates a non-statistically significant difference ($p > 0.05$), whereas a different letter above a histogram column indicates a statistically significant difference ($p < 0.05$).

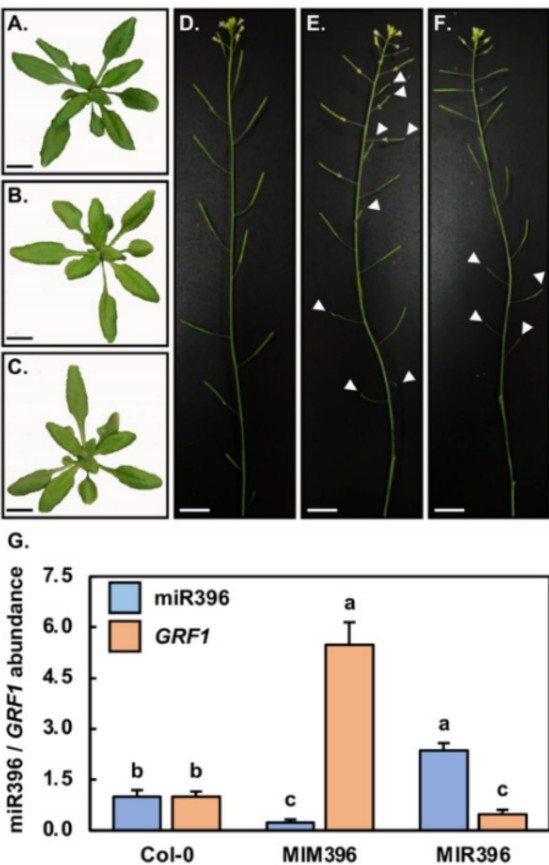

**Figure 1.** Phenotypic and molecular assessment of *MIM396* and *MIR396* transformant lines. Vegetative phenotypes displayed by 25-day-old soil grown Col-0 (**A**), *MIM396* (**B**), and *MIR396* (**C**) plants. Scale bar = 1.0 cm. Comparison of the reproductive development of 40-day-old soil-grown Col-0 (**D**), *MIM396* (**E**), and *MIR396* (**F**) plants. Scale bar = 2.0 cm. (**G**) RT-qPCR assessment of the abundance of miR396 and the expression of one of the primary target genes of miR396-directed expression regulation in *Arabidopsis*, *GRF1*, in the rosette leaves of 25-day-old soil grown Col-0, *MIM396*, and *MIR396* plants. Error bars represent the standard deviation of four biological replicates. Statistical data were analyzed using one-way ANOVA and Tukey's *post hoc* tests. Statistically significant differences indicated by a different letter ($p < 0.05$) above a histogram column.

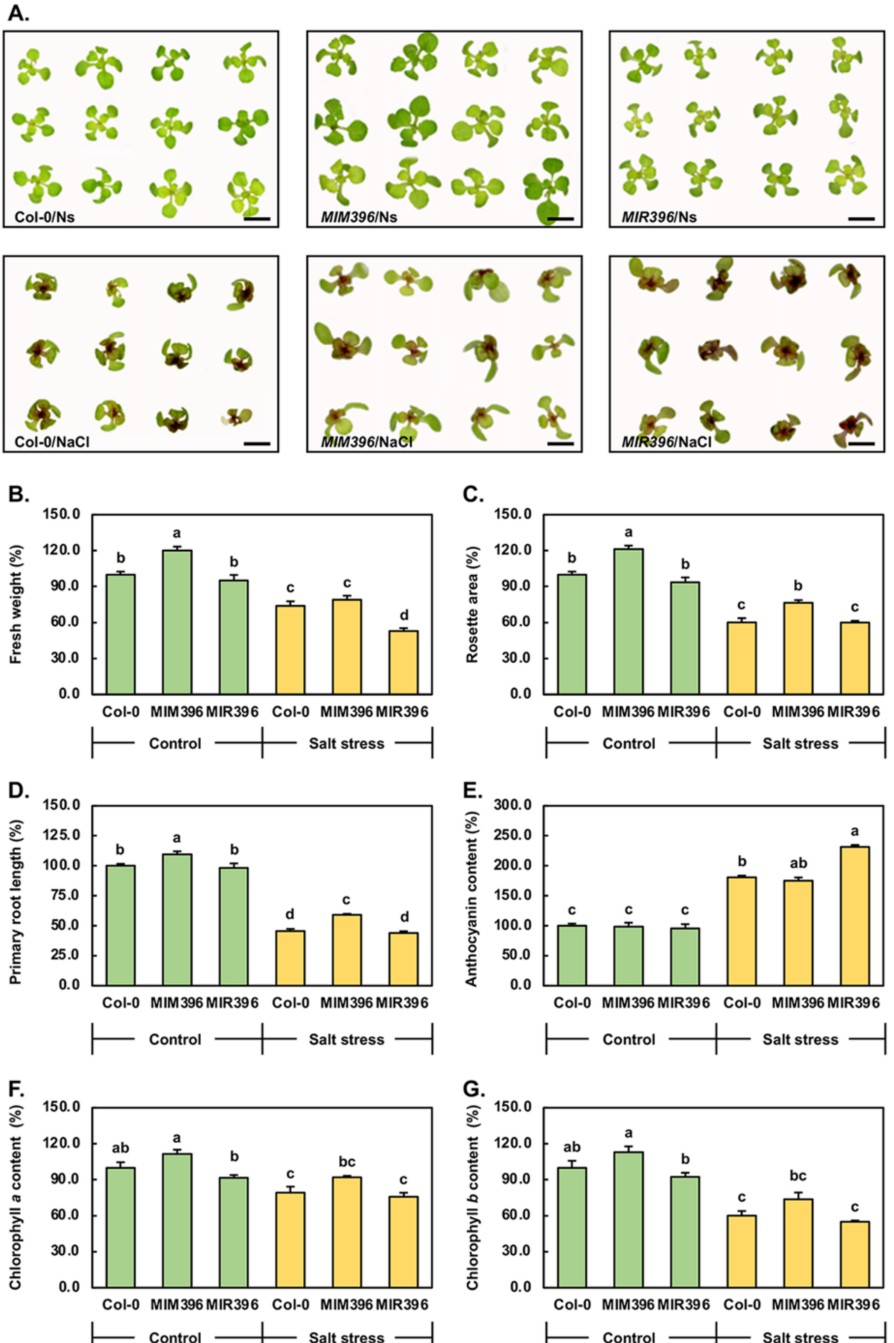

**Figure 2.** Phenotypic and physiological assessment of control and salt-stressed Col-0, *MIM396*, and *MIR396* plants. (**A**) Phenotypes displayed by 15-day-old control (top panels) and salt-stressed (bottom panels) Col-0, *MIM396* and *MIR396* plants. Scale bar = 1.0 cm. Quantification of the phenotypic parameters of (**B**) fresh weight (mg), (**C**) rosette area (mm$^2$), and (**D**) primary root length (mm) for control and salt-stressed 15-day-old Col-0, *MIM396* and *MIR396* plants. Quantification of the physiological parameters of (**E**) anthocyanin content (μg/g FW), as well as (**F**) chlorophyll *a* and (**G**) chlorophyll *b* abundance (mg/g FW), for control and salt-stressed 15-day-old Col-0, *MIM396* and *MIR396* plants. All phenotypic and physiological metrics are presented as a percentage (%) post comparison to the respective values obtained for Col-0/Ns plants, which were assigned a value of 100% for each assessed growth characteristic. (**B–G**) Error bars represent the standard deviation of four biological replicates. The statistical data were analyzed using one-way ANOVA and Tukey's *post hoc* tests with statistically significant differences denoted by a different letter ($p < 0.05$) above a histogram column.

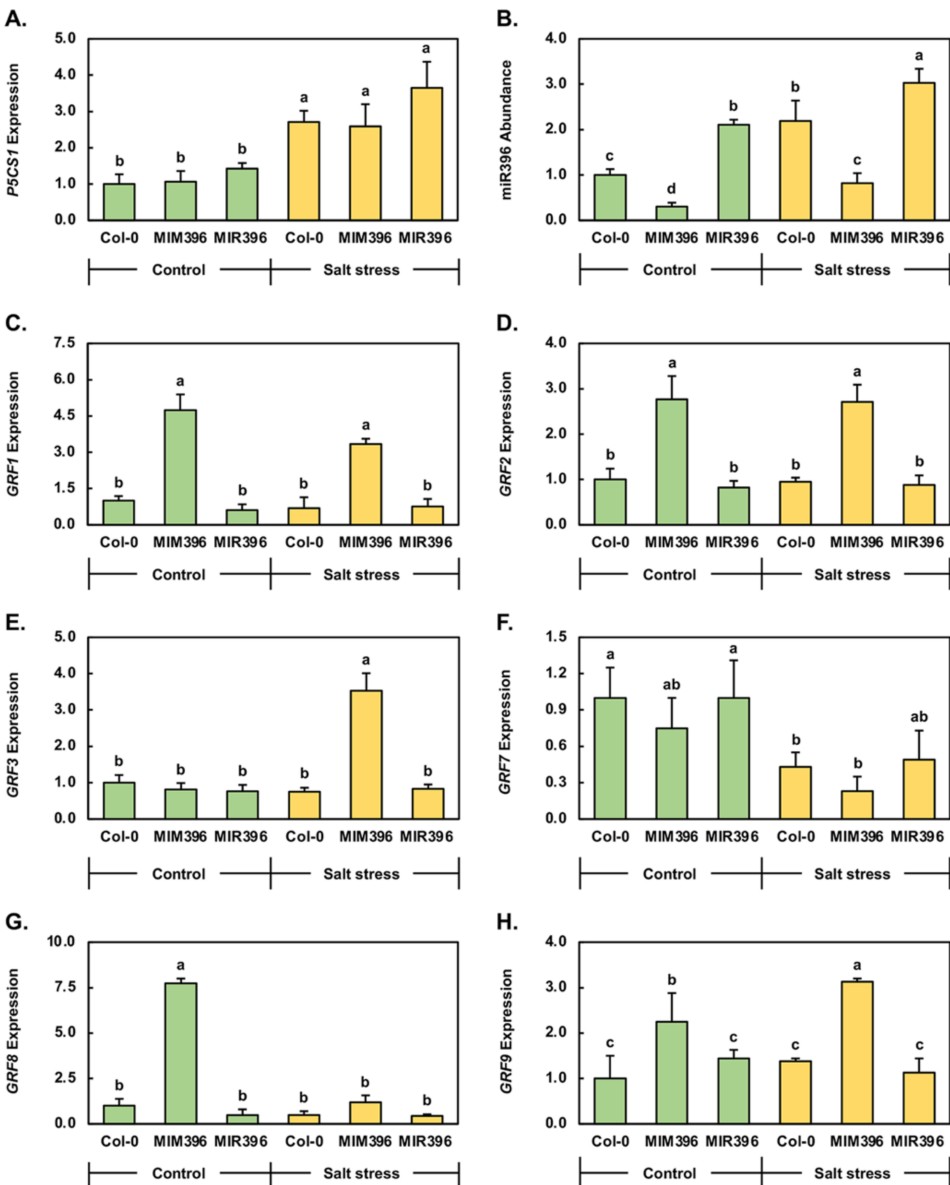

**Figure 3.** Molecular profiling of the miR396/*GRF* expression module in 15-day-old control and salt-stressed Col-0, *MIM396*, and *MIR396* plants. (**A**) RT-qPCR assessment of the expression of the *Arabidopsis* stress-responsive gene *P5CS1* in control and salt-stressed Col-0, *MIM396*, and *MIR396* plants. (**B**) Quantification of miR396 abundance by RT-qPCR in control and salt-stressed Col-0, *MIM396*, and *MIR396* plants. RT-qPCR quantification of the level of expression of the six *Arabidopsis* GRF gene family members known to form posttranscriptional targets of miR396-directed expression regulation, including *GRF1* (**C**), *GRF2* (**D**), *GRF3* (**E**), *GRF7* (**F**), *GRF8* (**G**), and *GRF9* (**H**) in Col-0/Ns, *MIM396*/Ns, *MIR396*/Ns, Col-0/NaCl, *MIM396*/NaCl, and *MIR396*/NaCl plants. (**A**–**H**) Error bars represent the standard deviation of four biological replicates. All statistical data were analyzed via the use of the one-way ANOVA and Tukey's *post hoc* tests, and a statistically significant difference ($p < 0.05$) is represented by a different letter above a histogram column.

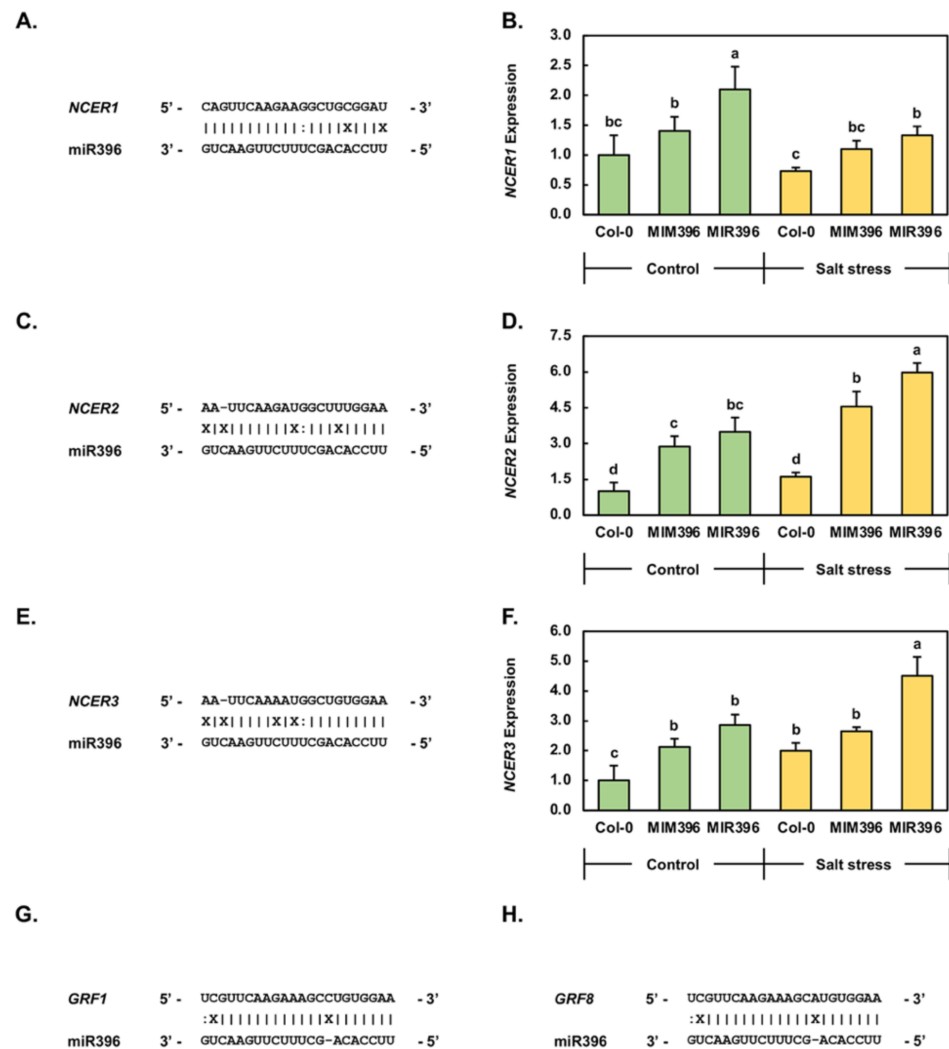

**Figure 4.** RT-qPCR assessment of *NCER1*, *NCER2*, and *NCER3* expression in control and salt-stressed Col-0, *MIM396*, and *MIR396* plants. Base pair predictions that putatively form between the miR396a sRNA and the *NCER1* (**A**), *NCER2* (**C**), and *NCER3* (**E**) transcripts. RT-qPCR analysis of *NCER1* (**B**), *NCER2* (**D**), and *NCER3* (**F**) gene expression in Col-0/Ns, *MIM396*/Ns, *MIR396*/Ns, Col-0/NaCl, *MIM396*/NaCl, and *MIR396*/NaCl plants. Base pairing which forms between miR396 and its two well-characterized target genes (**G**) *GRF1* and (**H**) *GRF8*. (**B,D,F**) Error bars represent the standard deviation of four biological replicates. The statistical data were analyzed using the one-way ANOVA and Tukey's *post hoc* tests, and a statistically significant difference is represented by a different letter ($p < 0.05$) above each histogram column.

## 3. Results

### 3.1. Phenotypic and Molecular Assessment of the MIM396 and MIR396 Transformant Lines

Two *Arabidopsis* transformant lines, termed *MIM396* and *MIR396* plants, were selected for further phenotypic and molecular assessment from the two newly generated transformant populations produced via a standard *Agrobacterium*-mediated approach [48]. More specifically, the *MIM396* transformant line was developed to have reduced miR396 abundance via the introduction and constitutive expression of an endogenous miRNA target mimicry (eTM) transgene specific to miR396 [46,47]. In contrast, the *MIR396* transformant line was generated to have elevated miR396 abundance via the introduction and constitutive expression of the *Arabidopsis PRE-MIR396A* precursor transcript, an approach which has been successfully applied previously [12,14,54–56].

Comparison of Figure 1A,B shows that the constitutive expression of the miR396-specific eTM transgene did not greatly alter the developmental trajectory of the *MIM396* transformant line during vegetative development with the rosette of a 25-day-old soil-grown *MIM396* plant (Figure 1B) largely indistinguishable to that of a soil-grown Col-0 plant of the same age (Figure 1A). However, although the size and shape of *MIM396* rosette leaves matched those of Col-0 plants, the rate of *MIM396* vegetative development was reduced with only 8 to 9 '*true*' leaves forming the rosette of 25-day-old *MIM396* plants (Figure 1B), compared to the 11 to 12 true leaves which formed the rosette of a 25-day-old Col-0 plant (Figure 1A). This represented a 26% reduction in the rate of rosette leaf emergence for the *MIM396* transformant line compared to wild-type *Arabidopsis*. An eTM transgene-based approach has been used previously in *Arabidopsis* [57,58], rice [59], and tomato [60] to sequester miR396 function; however, the rate of vegetative development progression was not reported in these previous studies. In *Arabidopsis* transformant lines expressing a miR396-specific eTM transgene, Casadevall et al. [57] showed that eTM rosette leaf development matched that of unmodified *Arabidopsis* plants. In tomato, however, the constitutive expression of an miR396-specific eTM was demonstrated by Cao et al. [60] to result in the development of leaves and other vegetative organs of reduced size due to a decrease in vegetative tissue cell size. As reported for the *MIM396* transformant line (Figure 1B), Figure 1C shows that the size and shape of *MIR396* rosette leaves were identical to those of Col-0 plants of the same age (Figure 1A); however, vegetative development was again observed to be delayed in the *MIR396* transformant line. More specifically, at 25 days of age, 9 to 10 true leaves formed the *MIR396* rosette compared to the 11 to 12 true leaves that formed the rosette of a 25-day-old soil-grown Col-0 plant (Figure 1A,C). This represented a slightly milder 17% reduction in the rate of rosette leaf formation in this transformant line, compared to the 26% reduction determined for the *MIM396* transformant line (Figure 1B,C). The unaltered architecture of *MIR396* rosette leaves contrasts with the previously reported findings in [8,12,54–56], which collectively demonstrated that the transgene-mediated overaccumulation of miR396 results in the production of rosette leaves that are shorter and narrower than those of Col-0 plants.

In contrast to the wild-type-like vegetative development of *MIM396* plants (Figure 1B), the reproductive development of this transformant line was severely compromised (Figure 1D,E). More specifically, siliques expanded to a uniform length regardless of their position on the Col-0 primary inflorescence (Figure 1D). In *MIM396* plants, however, all siliques expanded to a shorter length; furthermore, approximately 40–50% of *MIM396* siliques were greatly reduced in their overall length or failed to properly expand at all (Figure 1E; white triangles). Although not quantified, it is important to note that, upon full maturity, the shorter-length siliques of *MIM396* plants housed reduced seed numbers compared to the longer length siliques of Col-0 plants. An eTM approach has been used previously in *Arabidopsis* [57,58], rice [59], and tomato [60] to study the consequence of repressing miR396 accumulation. Of these studies [57–60], only the Cao et al. [60] study reported on the consequences to reproductive development of repressing miR396 abundance. More specifically, the generated miR396-specific eTM tomato transformant lines were demonstrated to have a reduced flower size, although the size of the resulting fruit was increased [60]. As reported for *MIM396* plants, the constitutive expression of the *PRE-MIR396A* precursor transcript negatively impacted the reproductive development of the *MIR396* transformant line (Figure 1F). Specifically, compared to 40-day-old soil-grown Col-0 plants (Figure 1D), silique length was reduced along the entire *MIR396* primary inflorescence with the elongation of siliques that formed from the proximal region of the inflorescence stem of *MIR396* plants (Figure 1F; white triangles), repressed to a greater degree than those that formed from the distal region of the *MIR396* inflorescence stem neighboring the terminal floral bud (Figure 1F). Although not quantified, reduced silique length was determined to result in each *MIR396* silique harboring a reduced number of seeds compared to the number of seeds housed by the longer-length siliques of Col-0 plants. The overexpression of either *Arabidopsis* miR396 precursor transcript, *PRE-MIR396A* or *PRE-MIR396B*, has been demon-

strated previously to negatively impact *Arabidopsis* reproductive development [8,13,14], with Liang et al. [14] going on to demonstrate that defective pistil development in their miR396 overexpression lines led to the observed reduction in seed number and, therefore, reduced silique elongation.

To demonstrate that the developmental phenotypes displayed by *MIM396* and *MIR396* plants resulted from the introduced molecular modifications, we next quantified miR396 abundance in the rosette leaves of 25-day-old soil-grown *MIM396* and *MIR396* plants via quantitative reverse transcriptase polymerase chain reaction (RT-qPCR) assessment. RT-qPCR revealed that, compared to the rosette leaves of 25-day-old soil-grown Col-0 plants, miR396 abundance was significantly reduced by 4.2-fold in *MIM396* plants (Figure 1G). RT-qPCR was next applied to confirm whether reduced miR396 abundance resulted in defective *GRF* target gene expression regulation in the *MIM396* transformant. Accordingly, RT-qPCR showed that *GRF1* expression was significantly elevated by 5.5-fold in *MIM396* rosette leaves (Figure 1G). In the *MIR396* transformant line, RT-qPCR revealed that miR396 abundance was significantly elevated by 2.4-fold compared to its abundance in Col-0 rosette leaves (Figure 1G). Furthermore, RT-qPCR subsequently showed that *GRF1* expression was reduced by 2.1-fold in the rosette leaves of *MIR396* plants (Figure 1G) in response to elevated miR396 abundance. Post quantification of miR396 abundance and *GRF1* target gene expression (Figure 1G), we suggest that the difference in the vegetative development of the *MIR396* transformant line generated in this study, compared to those lines produced in previous studies [8,12,54,55], is the result of the degree to which the miR396 sRNA overaccumulates, and therefore, the extent to which *GRF* target gene expression is repressed in each overexpression line. For example, Rodriguez et al. [54] demonstrated that, in their *MIR396* transformant line which displayed severely impacted vegetative development, miR396 accumulated to a level that was almost 10-fold higher than that of wild-type *Arabidopsis*. In contrast, the *MIR396* transformant line selected here for subsequent analysis was revealed via RT-qPCR to only have a moderate 2.4-fold enhancement in miR396 abundance (Figure 1G).

*3.2. Phenotypic and Physiological Analysis of 15-Day-Old Control and Salt-Stressed Col-0, MIM396, and MIR396 Plants*

Having previously demonstrated that miR396 abundance is altered in 15-day-old Col-0 whole seedlings following their cultivation for a 7-day period in the presence of 150 mM NaCl [19], 8-day-old Col-0, *MIM396*, and *MIR396* seedlings were exposed to the same salt stress treatment regime to further characterize the role played by the miR396/*GRF* expression module in the molecular response of *Arabidopsis* to this form of abiotic stress. Figure 2A clearly shows that the rosette leaf architecture of 15-day-old *MIM396* and *MIR396* plants remained the same as that of Col-0 plants of an equivalent age when all three *Arabidopsis* lines were cultivated on standard growth media (referred to as Col-0/Ns, *MIM396*/Ns, and *MIR396*/Ns plants herein). Although the architecture of *MIM396*/Ns rosette leaves matched that of Col-0/Ns plants, the overall development of the *MIM396*/Ns rosette appeared to be promoted (Figure 2A). In contrast, *MIR396*/Ns rosette development appeared to be mildly repressed compared to Col-0/Ns rosette development (Figure 2A). Figure 2A (bottom panels) further shows that the cultivation of 8-day-old Col-0, *MIM396*, and *MIR396* plants on *Arabidopsis* growth media supplemented with 150 mM NaCl for a 7-day period negatively impacted the developmental progression of all three plant lines (termed Col-0/NaCl, *MIM396*/NaCl, and *MIR396*/NaCl plants herein). More specifically, the continued expansion of rosette leaves and the elongation of rosette leaf petioles of Col-0/NaCl, *MIM396*/NaCl, and *MIR396*/NaCl plants was completely inhibited. In addition, the distal tips of Col-0/NaCl, *MIM396*/NaCl, and *MIR396*/NaCl rosette leaves curled downward toward the growth medium, and the uniform healthy green coloration displayed by Col-0/Ns, *MIM396*/Ns, and *MIR396*/Ns plants was replaced by a pale green-to-yellow coloration in the aerial tissues of Col-0/NaCl, *MIM396*/NaCl, and *MIR396*/NaCl plants. A brown-colored pigment was also observed to accumulate to a high abundance in the region surrounding the shoot apical meristem of salt-stressed Col-0, *MIM396* and

*MIR396* plants with the brown pigment accumulating to its highest abundance in the petioles and proximal leaf blade region of either emerging or fully emerged rosette leaves, or throughout the entire blade of newly emerged rosette leaves (Figure 2A).

To further assess the consequence of the molecular modifications made to the miR396/*GRF* expression module, the phenotypic parameters of (1) fresh weight, (2) rosette area, and (3) primary root length, together with the physiological parameters of the abundance of the plant pigments (4) anthocyanin, (5) chlorophyll *a*, and (6) chlorophyll *b*, were quantified (Figure 2B–G). The average fresh weight of a Col-0/Ns plant was 28.1 mg, and, as indicated by Figure 2A, the average fresh weight of an *MIM396*/Ns plant was determined to be significantly heavier by 20.2% at 33.8 mg (Figure 2B). The Figure 2A observations were further confirmed by the average fresh weight determination for a *MIR396*/Ns plant; specifically, at 26.7 mg, the average fresh weight of a *MIR396*/Ns plant was reduced by 5.0% compared to that of a Col-0/Ns plant. Via comparison to the control grown counterpart of each assessed plant line, the 7-day salt stress treatment regime was determined to reduce the fresh weight of Col-0/NaCl, *MIM396*/NaCl and *MIR396*/NaCl plants by 26.1, 41.2 and 42.2%, respectively (Figure 2B); this finding strongly inferred that both transformant lines were more sensitive to the imposed stress than were unmodified Col-0 plants. The average area of the rosette of a Col-0/Ns plant was 18.2 mm$^2$. In comparison, the average area of the rosette of *MIM396*/Ns and *MIR396*/Ns plants was 22.1 and 17.0 mm$^2$, representing a statistically significant increase of 21.3% and an insignificant and mild decrease of 6.5%, respectively (Figure 2C). The 7-day cultivation period in the presence of 150 mM NaCl was next revealed to significantly reduce the rosette area of a Col-0/NaCl plant by 39.9% to 10.9 mm$^2$. Similarly, compared to their control grown counterparts, the rosette areas of *MIM396*/NaCl and *MIR396*/NaCl plants were also determined to be significantly reduced by 44.8 and 33.5% to 16.9 and 10.2 mm$^2$, respectively (Figure 2C). Considering that altered root architecture is a well-documented phenotypic response of a plant to salt stress [61,62], the primary root length of control and salt-stressed Col-0, *MIM396*, and *MIR396* plants was compared (Figure 2D). This analysis showed that the average primary root length of a Col-0/NaCl plant was significantly reduced by 54.4% to 22.2 mm from the primary root length of 48.6 mm for a Col-0/Ns plant. The primary root lengths of *MIM396*/NaCl and *MIR396*/NaCl plants were next determined to be reduced by a similar degree. Specifically, compared to primary root lengths of 53.1 and 47.7 mm for *MIM396*/Ns and *MIR396*/Ns plants, the primary root lengths of *MIM396*/NaCl and *MIR396*/NaCl plants were reduced by 50.2 and 54.1% to 26.5 and 26.7 mm, respectively (Figure 2D). When taken together, assessment of this phenotypic parameter revealed that the development of the root system of all three *Arabidopsis* lines was negatively impacted to a significant, yet similar degree by the imposed salt stress treatment regime.

The physiological parameters of anthocyanin content and the abundance of chlorophyll *a* and *b* of control and salt-stressed Col-0, *MIM396*, and *MIR396* plants were assessed next. Spectrophotometry revealed the average anthocyanin content of a Col-0/Ns plant to be 2.2 µg/g FW. In comparison, the anthocyanin content of a Col-0/NaCl plant was 4.0 µg/g FW to represent an 80.8% elevation in the abundance of this flavonoid pigment (Figure 2E). The stress regime was next revealed to elevate the content of anthocyanin to a similar degree in the *MIM396* transformant line (Figure 2E). More specifically, the content of anthocyanin in an *MIM396*/NaCl plant was elevated by 76.9% to 4.3 µg/g FW, from 2.4 µg/g FW for an *MIM396*/Ns plant. In contrast, although the anthocyanin content of a *MIR396*/Ns plant was revealed to be highly similar to that of a Col-0/Ns plant at 2.1 µg/g FW, the anthocyanin content was revealed to be increased by a much greater degree, elevated by 135.9% to 4.9 µg/g FW, in the *MIR396*/NaCl sample (Figure 2E). Spectrophotometry was further used to quantify the abundance of the two primary photosynthetic pigments, chlorophyll *a* (Figure 2F) and *b* (Figure 2G), in control and salt-stressed Col-0, *MIM396*, and *MIR396* plants. The chlorophyll *a* content of Col-0/Ns, *MIM396*/Ns and *MIR396*/Ns plants was revealed to be 0.73, 0.80, and 0.69 mg/g FW, respectively. This represented a significant increase (elevated by 9.6%) in chlorophyll *a* content in *MIM396*/Ns

plants and a mild and insignificant reduction (decreased by 5.5%) in *MIR396*/Ns plants compared to that of Col-0/Ns plants (Figure 2F). It was next shown that the imposed salt stress treatment regime reduced the abundance of chlorophyll *a* by a similar degree, specifically, reductions of 21.8, 18.1 and 19.8% in Col-0/NaCl, *MIM396*/NaCl and *MIR396*/NaCl plants, respectively (Figure 2F). The chlorophyll *b* content of Col-0/Ns, *MIM396*/Ns, and *MIR396*/Ns plants was 0.18, 0.21, and 0.17 mg/g FW, respectively. This analysis showed that, when compared to the Col-0/Ns sample, the content of this primary photosynthetic pigment was significantly elevated by 16.7% in *MIM396*/Ns plants and mildly reduced by 5.6% in *MIR396*/Ns plants (Figure 2G). Spectrophotometry subsequently revealed that the degree of reduction in the content of chlorophyll *b* (Figure 2G) was considerably more than that of chlorophyll *a* (Figure 2F) following the application of the imposed stress. More specifically, compared to the control grown counterpart of each plant line, chlorophyll *b* abundance was reduced by 40.0, 36.8 and 37.5% in Col-0/NaCl, *MIM396*/NaCl and *MIR396*/NaCl seedlings, respectively (Figure 2G).

*3.3. Molecular Profiling of the miR396/GRF Expression Module in 15-Day-Old Control and Salt-Stressed Col-0, MIM396, and MIR396 Plants*

On the basis of our previous finding [19] that, in 15-day-old wild-type *Arabidopsis* plants, miR396 is responsive to salt stress, we further assessed this initial observation via first quantifying the level of expression of the *Arabidopsis* stress-responsive gene, *Δ1-PYRROLINE-5-CARBOXYLATE-SYNTHETASE1* (*P5CS1*; *AT2G39800*). This approach was applied to show that the three analyzed *Arabidopsis* lines were indeed experiencing stress at the molecular level following the application of the 7-day salt stress treatment regime. Compared to its level of expression in Col-0/Ns plants, RT-qPCR revealed that the abundance of the *P5CS1* transcript was mildly elevated by 1.1- and 1.4-fold in *MIM396*/Ns and *MIR396*/Ns plants, respectively (Figure 3A). When compared to its level of expression in the control grown counterpart of each *Arabidopsis* line, RT-qPCR subsequently showed that *P5CS1* expression was significantly elevated by 2.7-, 2.4-, and 2.6-fold in Col-0/NaCl, *MIM396*/NaCl and *MIR396*/NaCl plants, respectively (Figure 3A). This result readily showed that each *Arabidopsis* line was experiencing stress at the molecular level following the application of the imposed stress treatment regime. Post the demonstration that *P5CS1* expression was induced by a similar degree in the three assessed *Arabidopsis* plant lines following the application of the 7-day salt stress treatment regime, RT-qPCR was next used to quantify miR396 levels in Col-0/NaCl, *MIM396*/NaCl, and *MIR396*/NaCl plants for comparison to its abundance in Col-0/Ns, *MIM396*/Ns, and *MIR396*/Ns plants. Compared to its abundance in Col-0/Ns plants, miR396 levels were revealed to be significantly reduced and elevated by 3.3- and 2.1-fold in *MIM396*/Ns and *MIM396*/Ns plants, respectively (Figure 3B). RT-qPCR subsequently showed that, compared to Col-0/Ns, *MIM396*/Ns, and *MIR396*/Ns plants, miR396 abundance was elevated by 2.2-, 2.7-, and 1.4-fold in Col-0/NaCl, *MIM396*/NaCl, and *MIR396*/ NaCl plants, respectively (Figure 3B). Elevated miR396 abundance in 15-day-old Col-0, *MIM396*, and *MIR396* plants following the application of the 7-day salt stress treatment regime confirmed our previous demonstration [19] that, in *Arabidopsis*, miR396 is responsive to this form of abiotic stress.

The significant 3.3-fold reduction in miR396 abundance in *MIM396*/Ns plants, compared to the Col-0/Ns sample (Figure 3B), was next revealed by RT-qPCR to result in a significant elevation in the level of expression of *GRF1* (Figure 3C), *GRF2* (Figure 3D), *GRF8* (Figure 3G), and *GRF9* (Figure 3H) by 4.7-, 2.8-, 7.8-, and 2.3-fold, respectively. In contrast to this finding was the documentation of 1.2- and 1.3-fold reductions in the level of expression of *GRF3* and *GRF7*, respectively (Figure 3E, 3F). In the *MIR396*/Ns sample, where miR396 levels were significantly elevated by 2.1-fold (Figure 3B), *GRF1* (Figure 3C), *GRF2* (Figure 3D), *GRF3* (Figure 3E), and *GRF8* (Figure 3G) expression was reduced by 1.6-, 1.2-, 1.3-, and 1.6-fold, respectively. The expression of *GRF7* however, remained unchanged in *MIR396*/Ns plants (Figure 3F), and furthermore, *GRF9* expression was revealed to be mildly elevated by 1.4-fold (Figure 3H).

The consequence of altered miR396 abundance in salt-stressed Col-0, *MIM396*, and *MIR396* plants was next assessed via RT-qPCR. This approach showed that in response to the 2.2-fold elevation in miR396 abundance in Col-0/NaCl plants, *GRF1*, *GRF2*, *GRF3*, *GRF7* and *GRF8* expression was reduced by 1.5-, 1.1-, 1.3-, 2.3-, and 2.0-fold, respectively (Figure 3C–G). In contrast to the decreased abundance of these five miR396 target transcripts, *GRF9* expression was elevated by 1.4-fold in Col-0/NaCl plants (Figure 3H). In response to the 2.7-fold elevation in miR396 abundance in *MIM396*/NaCl plants, compared to its level in *MIM396*/Ns plants (Figure 3B), *GRF1* transcript abundance was mildly reduced by 1.4-fold (Figure 3C), and the expression of *GRF7* (Figure 3F) and *GRF8* (Figure 3G) was significantly decreased by 3.3- and 6.5-fold, respectively. In contrast, *GRF2* transcript abundance remained unchanged across control and salt-stressed *MIM396* plants (Figure 3D). Furthermore, the expression of the *GRF3* and *GRF9* target transcripts was elevated by 4.4- and 1.4-fold, respectively (Figure 3E,H). Post exposure of the miR396 overexpression line to the 7-day salt stress treatment regime, miR396 abundance was demonstrated via RT-qPCR to be further elevated by 1.4-fold from its already elevated abundance in *MIR396*/Ns plants (Figure 3B). Interestingly, this mild degree of additional elevation to miR396 abundance was revealed to result in mild and insignificant 1.1- to 1.2-fold increases in the expression of *GRF1*, *GRF2*, and *GRF3* in *MIR396*/NaCl plants, compared to their respective expression levels in *MIR396*/Ns plants (Figure 3C–E). However, the remaining three miR396 target transcripts assessed here, including *GRF7*, *GRF8*, and *GRF9*, were determined to have reduced levels of expression in the *MIR396*/NaCl sample compared to the *MIR396*/Ns sample. More specifically, in response to the 1.4-fold increase in miR396 abundance, *GRF7*, *GRF8*, and *GRF9* expression was decreased by 1.6-, 1.4-, and 1.3-fold, respectively (Figure 3F–H).

*3.4. Molecular Profiling of the Expression Trends of Three Additional Putative Target Genes of miR396-Directed Expression Regulation in Col-0, MIM396, and MIR396 Plants*

In addition to the six members of the *Arabidopsis GRF* gene family forming well characterized target genes of miR396-directed expression regulation, three members of the *NEUTRAL/ALKALINE NONLYSOSOMAL CERAMIDASE-LIKE* (*NCER*) gene family, including *NCER1* (*AT5G58980*), *NCER2* (*AT1G07380*), and *NCER3* (*AT2G38010*), have also been identified as putative targets of miR396-directed expression regulation at the posttranscriptional level [63]. Figure 4A shows that the miR396a sequence (the mature miRNA sRNA processed from the *PRE-MIR396A* precursor transcript) harbors 2.5 base pair mismatches (with a G:U wobble pair classed as a 0.5 mismatch) with the *NCER1* mRNA at nucleotide (nt) positions 1, 5 and 10 from the 5′ terminal nt of the miR396a sRNA. Considering that RT-qPCR revealed miR396 abundance to be reduced by 3.3-fold in the *MIM396*/Ns sample (Figure 3B), the expression of a '*bona fide*' posttranscriptional target of miR396-directed regulation would be expected to be elevated in this transformant line. Accordingly, Figure 4B shows that the abundance of the *NCER1* transcript was mildly elevated by 1.4-fold in *MIM396*/Ns plants. However, *NCER1* mRNA abundance was shown to be elevated by 2.0-fold in response to the 2.1-fold increase in the level of miR396 in *MIR396*/Ns plants (Figure 4B). The 7-day salt stress treatment regime was shown by RT-qPCR to elevate miR396 abundance by 2.2-, 2.4-, and 1.4-fold in Col-0/NaCl, *MIM396*/NaCl and *MIR396*/NaCl plants, respectively (Figure 3B). Accordingly, *NCER1* expression was determined to be reduced by 1.4-, 1.3-, and 1.6-fold in Col-0/NaCl, *MIM396*/NaCl and *MIR396*/NaCl plants, respectively, compared to its expression level in the control grown counterpart of each plant line (Figure 4B). Although largely reciprocal expression trends were constructed for the miR396 sRNA and the *NCER1* mRNA by RT-qPCR in control and salt-stressed Col-0, *MIM396*, and *MIR396* plants, the elevated abundance of the *NCER1* transcript in the *MIR396*/Ns sample (Figure 4B), where miR396 abundance was also elevated (Figure 3B), suggested that in 15-day-old *Arabidopsis* seedlings, *NCER1* does not form a posttranscriptional target of miR396-directed expression regulation. This suggestion was further supported by our inability to establish strong anticorrelation between the degree of miR396 elevation (Figure 3B) and the degree of reduced *NCER1* expression in

salt-stressed Col-0, *MIM396*, and *MIR396* samples (Figure 4B), when compared to the Col-0/Ns, *MIM396*/Ns, and *MIR396*/Ns samples, respectively.

The miR396a sequence contains 4.5 mismatched base pairings with its putative target site harbored by the *NCER2* mRNA, and these mismatched pairings were positioned at nt 6, 10 (G:U wobble pair), 11, 19, and 21 from the 5′ terminus of miR396a (Figure 4C). Compared to its expression level in Col-0/Ns plants, *NCER2* expression was significantly elevated by 2.8- and 3.5-fold in *MIM396*/Ns and *MIR396*/Ns plants (Figure 4D), in spite of miR396 abundance being significantly reduced and elevated in these two transformant lines, respectively (Figure 3B). In addition, *NCER2* expression was mildly increased in Col-0/NaCl, *MIM396*/NaCl, and *MIR396*/NaCl plants by 1.6- to 1.7-fold (Figure 4D), even though miR396 abundance was also elevated in these three *Arabidopsis* lines post their exposure to salt stress. As observed for *NCER2*, the hybridization product that potentially forms between miR396a and the *NCER3* transcript was determined to contain 4.5 base pair mismatches at nt positions 10 (G:U wobble pair), 11, 13, 19, and 21 from the 5′ terminal nt of the miR396a sRNA (Figure 4E). Compared to its level of expression in Col-0/Ns seedlings, *NCER3* expression was elevated by 2.8- and 3.5-fold in *MIM396*/Ns and *MIR396*/Ns plants, respectively (Figure 4F). RT-qPCR next revealed that in the Col-0/NaCl, *MIM396*/NaCl and *MIR396*/NaCl samples, *NCER3* expression was elevated by 2.0-, 1.2-, and 1.6-fold, respectively (Figure 4F). The abundance of miR396 was also elevated in Col-0, *MIM396*, and *MIR396* plants by the imposed salt stress treatment regime (Figure 3B); this finding indicated that like the *NCER1* (Figure 4B) and *NCER2* transcripts (Figure 4D), *NCER3* does not form a target of miR396-directed expression regulation at the posttranscriptional level in 15-day-old *Arabidopsis* whole seedlings.

## 4. Discussion

Considering the well-established role of the miR396/*GRF* expression module in plant development and the molecular response of a range of plant species to salt stress [1,4–6,10–15], here we report on the use of a molecular modification approach to generate two new *Arabidopsis* plant lines with altered miR396 abundance. This approach was undertaken to build on our original finding that in *Arabidopsis*, miR396 levels are altered in 15-day-old wild-type whole seedlings following their cultivation for a 7-day period in the presence of 150 mM NaCl [19]. Considering that the *MIM396* and *MIR396* transformant lines were molecularly modified to have reduced and elevated miR396 abundance, respectively, it was unsurprising to observe that these two *Arabidopsis* lines displayed opposing developmental trajectories to that of Col-0 seedlings. More specifically, when compared to Col-0/Ns plants, the assessed phenotypic parameters of fresh weight, rosette area, and primary root length were enhanced in *MIM396*/Ns plants (Figure 2B–D). In contrast, these three phenotypic parameters were all mildly reduced in control grown 15-day-old *MIR396* seedlings. As shown here in Figures 1B and 2A, no readily observable phenotypic differences were previously documented by the Casadevall et al. [57] study as part of the characterization of the vegetative development of their *Arabidopsis* transformant lines which expressed a miR396-specific eTM transgene. However, detailed quantitative assessments of developmental metrics such as fresh weight and rosette area were not performed by Casadevall et al. [57]; therefore, a direct comparison between the miR396-specific eTM transformant line generated in this study with those *Arabidopsis* lines generated previously [57], during the vegetative phase of *Arabidopsis* development, cannot be made. The constitutive expression of a miR396-specific eTM in tomato was however demonstrated by Cao et al. [60] to result in the development of leaves and other vegetative organs of reduced size due to a decrease in vegetative tissue cell size. The mild vegetative phenotype displayed by *Arabidopsis* lines expressing a miR396-specific eTM A,B and Figure 2A) [57], versus the severe vegetative phenotypes displayed by tomato transformants expressing an eTM transgene specific to miR396 [60], likely stems from the functional differences in the regulated target genes of each of the members of the *GRF* transcription factor gene family, which themselves form posttranscriptional targets of miR396-directed expression regulation in these two species.

When compared to nonmodified wild-type *Arabidopsis* plants, the mild degree of reduction in the phenotypic parameters of fresh weight, rosette area, and primary root length observed for *MIR396*/Ns plants (Figure 2B–D), versus the severe retardation of the vegetative development of *Arabidopsis* transformant lines molecularly modified by others previously to overaccumulate miR396 [8,12,54–56], is most likely the result of the degree of miR396 overexpression achieved. More specifically, in this study, miR396 abundance was enhanced by 2.4- and 2.1-fold in the $T_2$ (Figure 1G) and $T_3$ generation (Figure 3B), respectively, of the characterized *MIR396* transformant line. Accordingly, only a mild negative impact on vegetative development was observed for the *MIR396* transformant line when cultivated under standard growth conditions, either on soil (Figure 1A,C) or plant growth medium (Figure 2A–D). In direct contrast, the miR396 overexpression line generated as part of the Rodriguez et al. [54] study, an *Arabidopsis* line with severely impacted vegetative development as evidenced by shorter and more narrow rosette leaves, leading to an overall reduction in the size of the rosette, was demonstrated to have an approximate 10-fold higher abundance of miR396 than wild-type *Arabidopsis* plants. Therefore, the difference in the degree of negative developmental impact displayed by individual miR396 overexpression lines is likely a direct result of the level of enhanced target gene expression regulation (i.e., further repression of target gene expression) directed by miR396 in each transformant line.

The anthocyanin content of *MIM396*/Ns and *MIR396*/Ns seedlings remained largely unchanged to that of 15-day-old control grown Col-0 seedlings (Figure 2E). However, chlorophyll *a* and *b* abundance was elevated by 11.5% and 12.8%, respectively, in control grown *MIM396* seedlings, compared to the Col-0/Ns sample (Figure 2F,G). In contrast, chlorophyll *a* and *b* abundance was reduced by 8.2% and 7.6%, respectively, in the *MIR396*/Ns sample, compared to the Col-0/Ns sample (Figure 2E–G). A similar degree of response for each of the three assessed physiological parameters was observed for Col-0 and *MIM396* plants following the application of the salt stress treatment regime. Specifically, anthocyanin content was elevated by 80.8 and 76.9% in Col-0/NaCl and *MIM396*/NaCl plants, compared to Col-0/Ns and *MIM396*/Ns plants, respectively (Figure 2E). In addition, chlorophyll *a* abundance was reduced by 20.8 and 19.5% in salt-stressed Col-0 and *MIM396* plants (Figure 2F), and chlorophyll *b* levels were decreased by 40.0 and 39.1% in 15-day-old salt-stressed Col-0 and *MIM396* plants, when compared to the Col-0/Ns and *MIM396*/Ns samples, respectively (Figure 2G). The chlorophyll *a* and *b* content of salt-stressed *MIR396* plants was reduced to a slightly lesser degree than determined for either the Col-0 or the *MIM396* plant line post the salt stress treatment regime. More specifically, the level of chlorophyll *a* and *b* was reduced by 16.0 and 37.5% in the *MIR396*/NaCl sample compared to the *MIR396*/Ns sample (Figure 2F,G). Interestingly, it was previously reported that via its posttranscriptional regulation of the abundance of the *GRF7* and *GRF8* transcripts, miR396 may play an indirect role in regulating the biosynthesis of chlorophyll [64]. In direct contrast to the mild differences in the level of the two primary photosynthetic pigments of Col-0, *MIM396*, and *MIR396* plants to the applied stress, the content of anthocyanin in the *MIR396*/NaCl sample was dramatically elevated by 135.9% in the *MIR396*/NaCl sample (Figure 2E). Interestingly, the overexpression of miR396 in the hairy roots of red sage (*Salvia miltiorrhiza*), an important herbal plant in traditional Chinese medicine, was revealed to result in anthocyanin accumulating to a level 1.4-fold higher in the overexpression line than it did in unmodified red sage hairy roots [65]. Via the use of a transcriptomics approach, Zheng et al. [65] further revealed that the enhanced accumulation of anthocyanin in their miR396 overexpression lines was the result of elevated expression of the anthocyanin biosynthesis pathway genes, *CHALCONE SYNTHASE A2* (*CHS2*), *CHALCONE ISOMERASE2* (*CHI2*), and *LEUCOANTHOCYANIDIN HYDROLASE* (*LAR*), among others. Although *Arabidopsis* and red sage are evolutionary unrelated plant species, it seems likely that enhancement of the transcriptional activity of genes which encode the enzymes required for anthocyanin biosynthesis in *Arabidopsis* was the likely cause of the significantly

elevated abundance of this plant pigment in the *MIR396* transformant line post its exposure to salt stress.

The 2.2-, 2.7-, and 1.4-fold elevation in miR396 abundance in salt-stressed Col-0, *MIM396*, and *MIR396* plants, respectively (Figure 3B), readily confirmed our previous finding [19] that in 15-day-old *Arabidopsis* whole seedlings, miR396 is responsive to salt stress. Elevated miR396 abundance in response to salt stress has also been demonstrated in cotton, creeping bentgrass, switchgrass, tobacco, and tomato [23,25,26,44]. In contrast, the abundance of the miR396 sRNA is reduced in maize, rice, and *S. viridis* by salt stress [24,42,43]. However, when taken together, this large body of evidence strongly infers that altered miR396 abundance, and therefore, a change in the level of expression of the *GRF* target genes under miR396-directed posttranscriptional regulation, plays a central role in the adaptive response of this group of unrelated plant species to this form of abiotic stress. Further evidence of the central requirement of alteration to the miR396/*GRF* expression module as part of the molecular response of a plant to salt stress is provided by the heterologous expression studies of Gao et al. [24] and Chen et al. [26]. More specifically, the overexpression of a miR396 precursor transcript from rice in *Arabidopsis* was shown to increase the sensitivity of the generated transformant lines to salt stress [24], whereas the transformation of tobacco with a miR396 precursor transcript from tomato provided the resulting tobacco transformant lines with an enhanced tolerance to salt stress [26].

Adding further weight to the suggestion that alteration of the miR396/*GRF* expression module is central to the molecular response of Col-0 plants to this specific form of abiotic stress is provided in Figure 3C–H. More specifically, the expression of *GRF1*, *GRF2*, and *GRF3* was mildly to moderately reduced by 1.1- and 1.4-fold (Figure 3C–E); furthermore, the transcript abundance of *GRF7* and *GRF8* was significantly reduced by 2.3- and 2.0-fold, respectively (Figure 3F,G), in salt-stressed 15-day-old Col-0 whole seedlings. The expression anticorrelation observed for these five *GRF* target genes to that of their targeting sRNA, miR396, revealed that miR396 controls the expression of each of these *GRF* target genes via the canonical form of plant miRNA-directed target gene expression regulation, target transcript cleavage [66–68]. The expression of the sixth miR396 target gene analyzed in this study, *GRF9*, was determined to be elevated by 1.4-fold in response to the significantly increased abundance (2.2-fold) of miR396, and not decreased as expected in Col-0/NaCl plants, when compared to the Col-0/Ns sample (Figure 3H). Elevated *GRF9* expression in response to increased miR396 abundance in 15-day-old salt-stressed Col-0 whole seedlings could stem from (1) the abundance of this target transcript being regulated via the alternate mechanism of plant miRNA-directed RNA silencing, translational repression, with target transcript abundance shown to scale with that of the targeting miRNA when this form of silencing is in operation [66–68], or (2) the transcriptional activity of the *GRF9* locus being induced to a greater degree than was the expressional activity of miR396 encoding loci in *Arabidopsis*. In response to the significant 3.3-fold reduction in miR396 abundance in *MIM396*/Ns plants (Figure 3B), however, *GRF9* transcript abundance was significantly elevated by 2.3-fold (Figure 3H); this finding suggests that the *GRF9* transcript does indeed form a target of miR396-directed gene expression regulation at the posttranscriptional level, but that the degree to which the expression of the *GRF9* locus is induced could not be fully accounted for by the degree to which miR396 abundance was also elevated by the salt stress treatment regime applied in this study, thereby resulting in the observed increase in the level of both miR396 and *GRF9* in salt-stressed 15-day-old Col-0 whole seedlings.

Of the six *GRF* target genes analyzed, the expression of *GRF7* and *GRF8* was repressed to the greatest degree by the increased level of the targeting miRNA, miR396 (Figure 3B,F,G), in salt-stressed Col-0 seedlings. In *Arabidopsis*, the GRF7 transcription factor has been demonstrated to bind to specific *cis* elements in the promoter region of the *DREB2A* locus to repress *DREB2A* gene expression, which would in turn reduce the abundance of the DREB2A protein, which is associated with plant growth inhibition, yet provides a heightened tolerance to abiotic stress when it overaccumulates [17,69,70]. The *Arabidopsis grf7* single mutant also displays a heightened tolerance to osmotic stress which

likely results from the large cohort of stress-responsive genes with altered levels of expression in this mutant background [17,27]. Therefore, the observed 2.3-fold reduction in the level of *GRF7* gene expression in salt-stressed Col-0 whole seedlings (Figure 3F) may be required to modulate the transcriptional activity of the *DREB2A* locus [17,69,70] and that of the loci encoding the stress-responsive gene cohort with altered expression in the *grf7* single mutant, in an attempt by *Arabidopsis* to mount a molecular response to salt stress. Unlike GRF7, the regulatory role directed by the GRF8 transcription factor in *Arabidopsis* growth and development, or in the molecular response of *Arabidopsis* to environmental stress, has not yet been determined. However, the rice homolog of the *Arabidopsis* GRF8, *Os*GRF8, has been demonstrated to be involved in transcriptionally regulating the flavonoid biosynthesis pathway to mediate a degree of tolerance against the infestation of rice by the insect pathogen, brown planthopper (*Nilaparvata lugens*) [59]. This finding in rice is of high interest considering the significant 2.0-fold decrease in the abundance of the *GRF8* transcript in salt-stressed Col-0 whole seedlings, and it presents a new avenue for future investigation to establish the requirement of the miR396 sRNA and its *GRF8* target gene as part of the molecular response of *Arabidopsis* to a wide array of environmental stressors.

The overexpression of the *PRE-MIR396A* and *PRE-MIR396B* precursor transcripts to achieve elevated miR396 abundance has been demonstrated to result in the generated *Arabidopsis* transformant lines displaying an anticorrelation between elevated miR396 abundance and decreased expression of the three *NCER* gene family members which harbor putative miR396 target sites, namely, *NCER1*, *NCER2*, and *NCER3* [63]. Decreased *NCER1*, *NCER2* and *NCER3* expression in response to elevated miR396 accumulation putatively identified these three *NCER* gene family members as targets of miR396-directed expression regulation in *Arabidopsis* [63], in addition to the six members of the *GRF* transcription factor gene family [5–9,12–14]. In this study, RT-qPCR repeatedly failed to establish anticorrelation between the abundance of the miR396 sRNA (Figure 3B) and the *NCER1*, *NCER2*, and *NCER3* mRNA transcripts (Figure 4B,D,F) in control or salt-stressed Col-0, *MIM396*, and *MIR396* plants. These data indicate that the three assessed members of the *NCER* gene family do not form additional targets at the posttranscriptional level of miR396-directed expression regulation in 15-day-old *Arabidopsis* whole seedlings.

It has been previously shown [71–74] that the number of mismatches between a miRNA and a potentially recognized target sequence harbored by a putative miRNA target transcript, as well as the position of each mismatched base pair, is a crucial determinant of whether an *Arabidopsis* miRNA will exert a regulatory effect on the putatively targeted gene. The potential target site harbored by the *NCER1* transcript contains 2.5 mismatched base pairings to the miR396 sRNA (Figure 4A); an equivalent number of miRNA/target transcript base pair mismatches which are present between miR396 and its two well-characterized target transcripts, *GRF1* (Figure 4G) and *GRF8* (Figure 4H). In addition, the *GRF2* (Figure S1B) and *GRF7* (Figure S1D) transcripts also contain 2.5 mismatched base pairings with their targeting miRNA, miR396. All four of these *GRF* target transcripts harbor mismatched base pairings at nt positions 8, 20, and 21 (G:U wobble pair) from the 5' terminal nt of miR396a (Figure S1). In contrast, the base pair mismatches between the *NCER1* transcript and the miR396a sRNA occur at nt positions 1, 5, and 10 (G:U wobble pair) (Figure 4A). This finding, together with the expression trends presented in Figures 3 and 4, strongly suggest that although the *NCER1* transcript contains a putative miR396 target site, the position of the mismatched base pairs liberates *NCER1* from miR396-directed expression regulation at the posttranscriptional level. In contrast to *NCER1*, the putative miR396 target sites of the *NCER2* (Figure 4C) and *NCER3* transcripts (Figure 4E) contain 4.5 mismatched base pairings with the miR396 sRNA. The high number of base pair mismatches, together with the positioning of these mismatches, strongly suggests that the miR396 sRNA would not possess the degree of specificity required [71–74] to play a role in regulating the expression of either of these *NCER* gene family members in 15-day-old *Arabidopsis* whole seedlings.

## 5. Conclusions

Stemming from our initial finding that miR396 abundance is altered in 15-day-old wild-type *Arabidopsis* whole seedlings post their exposure to a 7-day salt stress treatment regime [19], the *MIM396* and *MIR396* transformant lines were generated to further characterize the requirement of altered miR396 abundance as part of the molecular response of *Arabidopsis* to this form of abiotic stress. Both the *MIM396* transformant line with reduced miR396 abundance, and the *MIR396* transformant line engineered to have an elevated miR396 level, displayed unique phenotypic and physiological responses to the imposed stress, phenotypic and physiological responses which differed from those displayed by unmodified 15-day-old wild-type *Arabidopsis* whole seedlings. The molecular profiling of salt-stressed Col-0, *MIM396*, and *MIR396* plants by RT-qPCR indicated that the unique phenotypic and physiological responses displayed by the two transformant lines likely stemmed from the individual expression trends constructed for each of the six *GRF* gene family members that form a target of miR396-directed expression regulation at the posttranscriptional level. RT-qPCR also revealed that, in 15-day-old *Arabidopsis* whole seedlings, the three assessed members of the *NCER* gene family do not form additional posttranscriptional targets of miR396-directed expression regulation. Considering that miR396 has been shown to be responsive to salt stress across a diverse range of unrelated plant species, together with the documented species-specific responses to the molecular alteration of the miR396/*GRF* expression module, the research presented here strongly suggests that a species-by-species approach is required in order to utilize a molecular modification approach in the future to engineer miR396-mediated resistance to this form of abiotic stress in agronomically important species such as rice, wheat, and maize.

**Supplementary Materials:** The following are available online at https://www.mdpi.com/article/10.3390/agronomy11091751/s1: Figure S1. Schematic representation of the base pairing between the miR396a sRNA and its six target genes which belong to the GRF transcription factor gene family; Table S1. Sequence information for all DNA oligonucleotides used in the RT-qPCR analyses presented in this study.

**Author Contributions:** Conceptualization, J.L.P., C.P.G. and A.L.E.; methodology, J.L.P. and J.M.O.; formal analysis, J.L.P., D.Q.N. and J.M.O.; investigation, J.L.P. and J.M.O.; resources, C.P.G., J.L.P. and A.L.E.; writing—original draft preparation, J.L.P., C.P.G. and A.L.E.; writing—review and editing, J.L.P., D.Q.N., J.M.O., C.P.G. and A.L.E.; supervision, C.P.G. and A.L.E. All authors read and agreed to the published version of the manuscript.

**Funding:** This research received no external funding.

**Data Availability Statement:** The data are available upon request to the authors, including the seeds from the generated *MIM396* and *MIR396* transformant lines.

**Acknowledgments:** The authors would like to thank fellow members of the Center for Plant Science for their guidance with plant growth care and RT-qPCR experiments.

**Conflicts of Interest:** The authors declare no conflict of interest.

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
