# Peer review of "Molecular Manipulation of the MiR396/GRF Expression Module Alters the Salt Stress Response of Arabidopsis thaliana"

_agronomy, doi:10.3390/agronomy11091751_

Round 1

Reviewer 1 Report

In this manuscript, the authors produced two transgenic Arabidopsis lines in which the levels of functional miR396 were increased or reduced, and they characterized the responses against the salt treatment and also determined the changes in the target mRNA levels with or without salt treatment. The results were clear and will be infomative for readers, especially for researchers dealing with salt tolerance. I recommend this manuscript to be published. However, several points should be addressed before publication, because several descriptions are obscure and have a room to be improved.

(1) The authors should provide a supplementary figure in which base pairing formed between miR396 and target genes, GRF1,2,3,7,8 and 9. It would be very helpful for readers to understand the discussion concerning to the base pairing miR396 and target genes.

(2) Line 659; Anthocyanin abundance was elevated in MIM396/Ns plants when compared to Col-0 seedings. 

Q: in the control group of Figure 2E, I cannot see the increased anthocyanin content in MIM396/Ns by comparison with salt-free Col-0 plants. The authors should describe the data based on the statistical differences. 

(3) Line 711 and following sentences: The expression of GRF1, 2, 3, 7, and 8 was reduced in response to elevated miR396 abundance in salt stressed Col-0 seedlings.

Q: I cannot see such reduction in GRF genes in the salt treatment. I compared Col-0 lane in control with Col-0 lane in salt stress. The alphabets on the bar which show the statistical analysis results were same (most of them are indicated as b in control and salt condition) except for the case of GRF7.

(4) Line 807: only NICER1 was identified as an additional posttranscriptional target of miR396.

Q; Why can the author identify the NICER1 not NICER2 and 3 as the target of miR396? I cannot totally recognize the differences among the exprssion of NICER1, 2, and 3 as shown in Figure 4. The mRNA abundances of these NICER genes were increased in both MIM396 and MIR396 transgenic plants. This characteristics was quite different from the expression of GRF1 in these transgenic lines. I cannot understand why NICER1 is regulated by miR396 from your data. 

(5) The last sentence in Conclusion

Q: The data shown in this manuscript showed the engineering of miR396-mediated resistance against salt stress was failed in Arabidopsis, because both transgenic lines showed weakened salt tolerance. As authors mentioned in line 637, functional of the miR396-regulated target genes of each members of the GRF gene family will be specifically established in each plant species. So the increased miR396 abundance caused increased salt tolerance in tobacco, but not in Arabidopsis. I recommend the conclusion of this part should be changed to more generalized contents. 

Author Response

Comments and Suggestions for Authors

In this manuscript, the authors produced two transgenic Arabidopsis lines in which the levels of functional miR396 were increased or reduced, and they characterized the responses against the salt treatment and also determined the changes in the target mRNA levels with or without salt treatment. The results were clear and will be informative for readers, especially for researchers dealing with salt tolerance. I recommend this manuscript to be published. However, several points should be addressed before publication, because several descriptions are obscure and have a room to be improved.

The authors would like to thank Reviewer #1 for their thorough and constructive review of our submitted manuscript. The authors would also like to take this opportunity to thank Reviewer #1 for recommending publication acceptance of our manuscript. We have attempted to address all concerns raised by Reviewer #1 in the revised version of our manuscript. Please see below our point-by-point responses to each of the concerns/comments raised by Reviewer #1.

Please also note: during the author revision process and the viewing of the various stages of the revised manuscript on either an Apple Mac, or a standard PC (different for each author), the formatting of the word file appears to have been altered. We have therefore supplied line numbers and page numbers for each specific correction made to the revised manuscript for ease of Reviewer reference.

(1) The authors should provide a supplementary figure in which base pairing formed between miR396 and target genes, GRF1,2,3,7,8 and 9. It would be very helpful for readers to understand the discussion concerning to the base pairing miR396 and target genes.

We thank Reviewer #1 for this helpful suggestion, and the authorship team have addressed this concern via the addition of Supplemental Materials Figure S1. Further, information relating to the inclusion of Figure S1 has been provided in the Supplementary Materials section of the revised manuscripts (lines 1017-1020, page 18/21 [on Apple Mac computer]; or lines 881-885, page 19-20/22 [on a PC]). We have also referenced Figure S1 in the revised Discussion section of the manuscript (please see lines 971-993, page 18/21 [on Apple Mac computer]; or lines 833-837, page 19/22 [on a PC]).

(2) Line 659; Anthocyanin abundance was elevated in MIM396/Ns plants when compared to Col-0 seedings. 

Q: in the control group of Figure 2E, I cannot see the increased anthocyanin content in MIM396/Ns by comparison with salt-free Col-0 plants. The authors should describe the data based on the statistical differences. 

The authorship team thank Reviewer #1 for identifying this oversight, an oversight which we do apologise for. To address this oversight, and to further correct the text surrounding this issue, we have reworded this entire section which now constitutes lines 659 to 719 on pages 15-16/21 of the revised manuscript ([on Apple Mac computer]; or lines 675-714, page 16/22 [on a PC]). We have also changed the order of references cited in this section from 65 to 64, and therefore, citation 64 has become citation 65 in the revised manuscript version.

(3) Line 711 and following sentences: The expression of GRF1, 2, 3, 7, and 8 was reduced in response to elevated miR396 abundance in salt stressed Col-0 seedlings.

Q: I cannot see such reduction in GRF genes in the salt treatment. I compared Col-0 lane in control with Col-0 lane in salt stress. The alphabets on the bar which show the statistical analysis results were same (most of them are indicated as b in control and salt condition) except for the case of GRF7.

We have reworded the text of the Discussion relating to the data presented in Figure 3C to 3G to address this concern of Reviewer #1. The expression of GRF1, GRF2 and GRF3 was mildly to moderately reduced in salt-stressed Col-0 plants, however; the degree of reduction was not determined to be statistically significant. The expression of GRF7 and GRF8 was also reduced in the Col-0/NaCl sample, and the documented decrease was determined to be statistically significant. Therefore, the text of the Discussion (lines 749-760 of page 16/21 [on an Apple Mac computer]; or lines lines 749-760, page 17/22 [on a PC]) has been altered to align with the statistical data more accurately.

(4) Line 807: only NICER1 was identified as an additional posttranscriptional target of miR396.

Q; Why can the author identify the NICER1 not NICER2 and 3 as the target of miR396? I cannot totally recognize the differences among the exprssion of NICER1, 2, and 3 as shown in Figure 4. The mRNA abundances of these NICER genes were increased in both MIM396 and MIR396 transgenic plants. This characteristics was quite different from the expression of GRF1 in these transgenic lines. I cannot understand why NICER1 is regulated by miR396 from your data.

The authors thank Reviewer #1 for raising this concern. In the revised version of our manuscript, we have extensively reworded the text of the Discussion section relating to the NCER expression trends in control grown and salt stressed Col-0, MIM396 and MIR396 plants. The included supplement Figure, Figure S1, has also been referred to in the revised text as part of the authorship team’s attempt to address this reviewer concern (please see lines 961-965 of page 17/21, and lines 966-995 of page 18/21 of the revised manuscript [on an Apple Mac computer]; or lines 810-824, page 18/22, and lines 830-853 of pages 18-19/22 [on a PC]).  

(5) The last sentence in Conclusion

Q: The data shown in this manuscript showed the engineering of miR396-mediated resistance against salt stress was failed in Arabidopsis, because both transgenic lines showed weakened salt tolerance. As authors mentioned in line 637, functional of the miR396-regulated target genes of each members of the GRF gene family will be specifically established in each plant species. So the increased miR396 abundance caused increased salt tolerance in tobacco, but not in Arabidopsis. I recommend the conclusion of this part should be changed to more generalized contents. 

To address this concern of the Reviewer, we have made numerous changes to the wording of most of the Conclusion section to attempt to make the overall wording of this section of the revised manuscript more generalized. Specifically, please see lines 1001 to 1014 of page 18/21 (when viewed on an Apple Mac computer; or, lines 861-880 of page 19/22 when viewed on a PC computer) of the revised manuscript for the wording changes made to this section of the manuscript to address this reviewer concern.

Reviewer 2 Report

Thank you for the opportunity to review this manuscript on the regulation of miR396 and its target genes in Arabidopsis during salt stress treatment. The study is built upon a previously reported induction of miR396 by salt stress, and the authors described the phenotypic and molecular responses of two Arabidopsis transpormant lines with altered miR396 expression 7 days after salt stress. The authors individually examined the change in expression of previously described targets including GRF and NCER genes.

The manuscript is well-written and easy to follow. While the authors have comprehensively described their results and compared with previous studies, I found the topic on miR396 was well-studied, and I failed to see the novelty in this study. The target genes included in this study have previously been described. The salt-stress responses of their different transformant lines are similar amongst each other. It would be helpful if the authors can improve the manuscript by highlighting the significance of their findings in comparison to previous studies in the area. However, I believe the study is done in a scientifically-sound fashion despite a few minor issues, and it certainly help to validate our current understanding on miR396 regulation in Arabidopsis. I therefore, would encourage the authors to work on the following questions regarding the manuscript prior to proceeding to publication.

  1. Why is there a focus on 15-day-old arabidopsis and salt stress response at 7 days? Is it an arbitrary choice? I wonder if the level of miR396 and its targets vary at different timepoints of the salt treatment?
  2. Following up on the previous question, I notice that in figure 1, the abundance of miR396 was differ in different magnitude in different transformants, in comparison to figure 3. Given that the plants used in figure 1 is 25-day-old, whereas figure 3 is 15-year-old, I wonder if this indicates changes in miR396 abundance at different ages, and would this change differ in different transformants ( i.e. age x mutation interaction effect). Would the authors see a different expression response on miR396 and its target genes when the treatment was done at a different age?
  3. With the use of 1-way ANOVA, the authors cannot separate the salt-treatment effect from the transformant effect. I'd suggest to use 2-way ANOVA to help the interpretation on the different effects.
  4. According to figure 3, it seems to me that certain GRF genes such as GRF2 and GRF1 are expressed in similar level in control and salt stress. Given that miR396 increase in salt stress, does this unresponsiveness suggest these GRF may not be targeted by miR396? I found that this observation contradict to your claim at line 713-714.
  5. What is the reason behind the selection of salt concentration for the salt treatment. Do you observe any differences in phenotype or gene expressions at a higher/lower level?
  6. Can the authors point out more specifically, the significance of their study in comparison to the previous miR396 studies done on Arabidopsis or other species of plants?
  7. Can the authors point out the linkage between the phenotypic changes and the target gene expression changes? For example, are the GRF genes and NCER genes related to the production of chlorophyll a and b or anthocyanin?
  8. Line 159: It should be MIM396 and MIR396 transformant, I believe? Please change the wording on this section to avoid self-plagiarism if similar method has been used for studying a different miRNA in another published study.

Author Response

Comments and Suggestions for Authors

Thank you for the opportunity to review this manuscript on the regulation of miR396 and its target genes in Arabidopsis during salt stress treatment. The study is built upon a previously reported induction of miR396 by salt stress, and the authors described the phenotypic and molecular responses of two Arabidopsis transpormant lines with altered miR396 expression 7 days after salt stress. The authors individually examined the change in expression of previously described targets including GRF and NCER genes.

The manuscript is well-written and easy to follow. While the authors have comprehensively described their results and compared with previous studies, I found the topic on miR396 was well-studied, and I failed to see the novelty in this study. The target genes included in this study have previously been described. The salt-stress responses of their different transformant lines are similar amongst each other. It would be helpful if the authors can improve the manuscript by highlighting the significance of their findings in comparison to previous studies in the area. However, I believe the study is done in a scientifically-sound fashion despite a few minor issues, and it certainly help to validate our current understanding on miR396 regulation in Arabidopsis. I therefore, would encourage the authors to work on the following questions regarding the manuscript prior to proceeding to publication.

The authors would like to take this opportunity to thank Reviewer #2 for their supportive assessment of our submitted manuscript. Please find below our point-by-point response to the minor concerns raised by Reviewer #2 which outline the specific changes that we have made to the revised manuscript to attempt to address the concerns of Reviewer #2 and to improve on our original submission.

Please also note: during the author revision process and the viewing of the various stages of the revised manuscript on either an Apple Mac, or a standard PC (different for each author), the formatting of the word file appears to have been altered. We have therefore supplied line numbers and page numbers for each specific correction made to the revised manuscript for ease of Reviewer reference.

  1. Why is there a focus on 15-day-old arabidopsis and salt stress response at 7 days? Is it an arbitrary choice? I wonder if the level of miR396 and its targets vary at different timepoints of the salt treatment?

In our original study published in the MDPI journal Plants in 2019 (PMID: 30857364), we studied three abiotic stresses (salt stress, drought stress and heat stress) in parallel with one another. As part of this initial study, a 7-day exposure period of 8-day-old Arabidopsis seedlings was identified as the most appropriate stage of Arabidopsis development where wild-type (Col-0) seedlings displayed a readily observable phenotypic and/or physiological response to all three forms of abiotic stress assessed in this study. Considering that miR396 was identified in our 2019 Plants study as a miRNA of interest which warranted further investigation, and that the molecularly modified plant lines with altered miR396 abundance which were generated for this present study, are based on the findings of our 2019 Plants study, the same stress conditions (i.e., a 7-day salt stress treatment period of 8-day-old Col-0, MIM396 and MIR396 seedlings) were applied here for consistency across these two closely related studies. Further, yes one would expect both the level of miR396 and that of its GRF target genes to continually change throughout Arabidopsis development considering that miR396-directed regulation of the expression of its GRF target genes has been repeatedly demonstrated previously to be essential for the correct development of Arabidopsis and other plant species. In addition, and as stated by Reviewer #2 in Point 2 below, demonstration of the degrees of difference in miR396 accumulation and/or GRF target gene expression already exist in our present study, that is: Figure 1G reports on miR396 abundance and GRF1 expression in the rosette leaves sampled from 25-day-old soil grown Col-0 plants, whereas Figures 3B and 3C report on miR396 abundance and GRF1 expression, in 15-day-old Col-0 whole seedlings, respectively. The different stage of vegetative development, and different growth conditions reported on Figures 1 and 3, clearly show the influence of growth stage, and growth condition, on miR396 abundance and the level of GRF1 expression.

  1. Following up on the previous question, I notice that in figure 1, the abundance of miR396 was differ in different magnitude in different transformants, in comparison to figure 3. Given that the plants used in figure 1 is 25-day-old, whereas figure 3 is 15-year-old, I wonder if this indicates changes in miR396 abundance at different ages, and would this change differ in different transformants ( i.e. age x mutation interaction effect). Would the authors see a different expression response on miR396 and its target genes when the treatment was done at a different age?

The differences in the abundance of miR396 in Figure 1G compared to its level in Figure 3B likely stems from a combination of four factors, namely; (1) the different ages of the plants at sampling (and therefore, a different stage of vegetative development) (as mentioned by Reviewer #2); (2) the different growth substrates that these two sets of plants were cultivated on (Figure 1G plants were cultivated on soil, versus Figure 3B plants which were cultivated on standard Arabidopsis growth media); (3) the differences in the tissues sampled for the RT-qPCR analyses presented in Figures 1G and 3B (Figure 1G were sampled from the rosette leaves of 25-day-old plants, versus Figure 3B which were sampled from 15-day-old whole seedlings (root and aerial tissues included)), and; (4) the Figure 1G analyses were performed on the T2 generation of MIM396 and MIR396 transformants, whereas the Figure 3B analyses report on the subsequent T3 generation of the two selected transformant lines. So yes, if the same stress treatment regime (a 7-day growth period in the presence of 150 mM NaCl), was applied to the Col-0, MIM396 and MIR396 plant lines at a later stage of development, different phenotypic, physiological, and molecular responses to the same stress treatment regime would be expected.

  1. With the use of 1-way ANOVA, the authors cannot separate the salt-treatment effect from the transformant effect. I'd suggest to use 2-way ANOVA to help the interpretation on the different effects.

The authors thank Reviewer #2 for this suggestion. However, the authors are of the opinion that the 1-way ANOVA used to statistically assess our data in the originally submitted manuscript version is the more appropriate form of statistical analysis of the reported data. Our reasoning: for the non-stressed samples, the values obtained for the MIM396/Ns and MIR396/Ns transformant lines are only compared to the corresponding value obtained for the Col-0/Ns sample, whereas for the salt-stressed samples, the Col-0/NaCl sample was only compared to the Col-0/Ns sample, the MIM396/NaCl sample was only compared to the MIM396/Ns sample, and the MIR396/NaCl sample was only compared to the MIR396/Ns. Therefore, on each occasion, only a single variable was assessed. Please also see similar studies where a 1-way ANOVA has been successfully applied:

  1. https://www.mdpi.com/1422-0067/20/1/153/htm
  2. https://www.mdpi.com/2073-4425/9/10/475/htm
  3. https://www.mdpi.com/2223-7747/10/3/452/htm

  1. According to figure 3, it seems to me that certain GRF genes such as GRF2 and GRF1 are expressed in similar level in control and salt stress. Given that miR396 increase in salt stress, does this unresponsiveness suggest these GRF may not be targeted by miR396? I found that this observation contradict to your claim at line 713-714.

The authors do not agree with this observation/statement made by Reviewer #2. More specifically, in salt-stressed Col-0 plants, the abundance of miR396 is elevated by 2.2-fold (Figure 3B), and accordingly the expression level of GRF1 is reduced by 31.0% (Figure 3C), GRF2 expression is reduced by 5.0% (Figure 3D), GRF3 expression is reduced by 25.0% (Figure 3E), GRF7 expression is reduced by 57.0% (Figure 3F), and the expression of GRF8 is reduced by 51.0% (Figure 3G). Similar to what is outlined in Points 1 and 3 above, differences to the degree of reduced GRF target gene expression resulting from salt-stress induced miR396 overaccumulation would likely stem from the degree of similarity/overlap of the expression domains of the targeting miRNA, miR396, and each of its GRF target genes. More specifically, considering that 15-day-old whole seedlings (including all root and aerial tissues) were sampled for the analyses presented in Figure 3, GRF target genes with a highly similar expression domain/s to the accumulation profile of miR396 would be expected to have large reductions to their level of expression (i.e., GRF7 and GRF8), whereas, GRF target genes with expression domains which only have a minor degree of overlap with the accumulation profile of miR396 would be expected to return only a mild to moderate degree of expression reduction (GRF1, GRF2 and GRF3) regardless of the degree to which the miR396 sRNA over-accumulates (i.e., miR396 can only regulate the expression of one of its target genes if both are expressed in the same tissue and/or cell type). We have also modified the text of lines 747 to 752 of the revised manuscript version to more accurately describe the expression changes documented for GRF1, GRF2, GRF3, GRF7 and GRF8 in salt-stressed Col-0 plants. We also provide a clear explanation as to why GRF9 expression is increased in salt-stressed Col-0 plants (Figure 3H) where miR396 accumulation was also determined to be increased (Figure 3B). Please see lines 756 to 924 of pages 16-17/21 (when viewed on an Apple Mac computer) (or lines 756-776 of pages 16-17/22 when viewed on a PC) of the revised manuscript for this explanation.

  1. What is the reason behind the selection of salt concentration for the salt treatment. Do you observe any differences in phenotype or gene expressions at a higher/lower level?

The concentration of the salt stress that we applied here (150 mM NaCl) was determined from our preliminary analyses as part of our original study published in Plants in 2019 (PMID: PMID: 30857364). More specifically, a 7-day cultivation period in the presence of 150 mM NaCl was determined to induce a moderate stress response by 15-day-old wild-type (Col-0) seedlings. Post application of the 7-day salt stress treatment regime, the fresh weight, rosette area and primary root length all showed readily observable reductions. In addition, anthocyanin accumulated to readily observable levels, and the healthy green colouration of the rosette leaves of control grown seedlings changed to a pale green to yellow colour in salt-stressed Col-0 seedlings. As part of these preliminary analyses, 75 mM NaCl was determined to only induce a very mild stress response in 15-day-old Col-0 seedlings, and a 7-day cultivation period on Arabidopsis growth medium supplemented with 250 mM NaCl induced a very severe stress response: a response which the authors were concerned would mask any ‘biologically reflective’ gene expression changes due to its severity. Please see our previously published studies (PMID: 33396498; PMID: 33114207) where we have used the same salt stress treatment regime to that applied in this study (a salt stress treatment regime determined suitable for 15-day-old Arabidopsis whole seedlings cultivated on standard Arabidopsis growth media (see PMID: 30857364).

  1. Can the authors point out more specifically, the significance of their study in comparison to the previous miR396 studies done on Arabidopsis or other species of plants?

  • Here we confirm the previous findings of Casadevall et al. (2013) (PMID: 19453503) who reported that the in planta expression of an eTM transgene specific to miR396 had no effect on vegetative development of Arabidopsis plants cultivated on soil.
  • However, this is study also forms the first report of the promoted development of 15-day-old Arabidopsis MIM396 plants during the early stage of their vegetative development: this too forms an interesting finding considering that the MIM396 line returns to the rate of wild-type Arabidopsis development at a subsequent stage of vegetative development (i.e., 25-day-old seedlings and beyond)
  • This study also represents the first report in Arabidopsis that the in planta expression of an eTM transgene specific to miR396 has a severe negative impact on the reproductive competence of This highly novel finding will be investigated further in the future by our investigative team.
  • Here we show that the lower level of miR396 overaccumulation achieved in this study, compared to previous studies, such as the study performed by Rodriguez et al., (2010) (PMID: 27794260), does not negatively impact Arabidopsis vegetative development. This too identifies the requirement of additional deconstruction of the miR396/GRF expression module to confirm the specific miR396 target genes with altered expression responsible for the difference in the vegetative phenotypes expressed between the lines generated here to those previously generated (Rodriguez et al., 2010) which achieved a higher degree of miR396 overaccumulation.
  • The heterologous expression of a rice MIR396 precursor transcript revealed a reduced tolerance to salt stress. We too, based on our phenotypic and physiological analyses presented here (Figure 2) show that miR396 overexpression results in decreased development progression (Figures 1 and 2) and an enhanced sensitivity to salt stress at the phenotypical and physiological levels (Figure 2).
  • This study also represents the first direct assessment in Arabidopsis of the role played by miR396 in the response of 15-day-old Arabidopsis whole seedlings to their exposure to a 7-day cultivation period in the presence of 150 mM NaCl.
  • This study also represents the first demonstration that in 15-day-old control grown or salt-stressed whole Arabidopsis seedlings, the previously identified NCER target genes, NCER1, NCER2 and NCER3, do not form posttranscriptional targets of miR396-directed expression regulation as previously reported by Goa et al., (2008) in Molecular Biology Reports.

  1. Can the authors point out the linkage between the phenotypic changes and the target gene expression changes? For example, are the GRF genes and NCER genes related to the production of chlorophyll a and b or anthocyanin?

Such examples of the linkage between altered GRF target gene expression and the differences in the assessed phenotypic and physiological parameters have already been made in the original manuscript version. Please see revised manuscript lines 734 to 737 (page 16/21) (or lines 697-700 of page 16/22 when viewed on a PC) for the text which was present in our original submission linking of the roles of miR396, GRF7 and GRF8 to chlorophyll biosynthesis, and similarly, please see revised manuscript lines 740 to 752 (page 16/21) (or lines 703-715 of page 16/22 when viewed on a PC) for the text which was present in our original submission linking altered miR396 abundance to the changed expression of genes encoding for anthocyanin biosynthesis enzymes. We also included information on the rice homolog of Arabidopsis GRF8, OsGRF8, which has been shown to play a role in transcriptionally regulating the flavonoid biosynthesis pathway in rice in response to pathogen attack (see revised manuscript lines 1003 to 1011, page 17/21 (when viewed on an Apple Mac computer) (or lines 793-801 of page 18/22 when viewed on a PC). It will therefore be of interest in the future to further characterise the roles played by specific members of the GRF transcription factor gene family, which also form targets of miR396-directed expression regulation, in the transcriptional regulation of the chlorophyll and flavonoid (including the anthocyanins) biosynthesis pathways as part of the response of Arabidopsis to this form abiotic stress.  

  1. Line 159: It should be MIM396 and MIR396 transformant, I believe? Please change the wording on this section to avoid self-plagiarism if similar method has been used for studying a different miRNA in another published study.

The authors kindly thank Reviewer #2 for identifying this mistake. We have corrected this mistake in the revised version of our originally submitted manuscript.